# DISENTANGLING PROPERTIES OF CONTRASTIVE METHODS

## ABSTRACT

Disentangled representation learning is an important topic in representation learning, since it not only allows the representation to be human interpretable, but it is also robust and benefits downstream task performance. Prior methods achieved initial successes on simplistic synthetic datasets but failed to scale to complex real-world datasets. Most of the previous methods adopt image generative models, such as GAN and VAE, to learn the disentangled representation. But we observe they are hard to learn disentangled representation on real-world images. Recently, self-supervised contrastive methods such as MoCo, SimCLR, and BYOL have achieved impressive performances on large-scale visual recognition tasks. In this paper, we explored the possibility of using contrastive methods to learn a disentangled representation, a discriminative approach that is drastically different from previous approaches. Surprisingly, we find that the contrastive method learns a disentangled representation with only minor modifications. The contrastively learned representation satisfies a "group disentanglement" property, which is a relaxed version of the original disentanglement property. This relaxation might be useful for scaling disentanglement learning to large and complex datasets. We further find contrastive methods achieve state-of-thet-art disentanglement performance on several widely used benchmarks, such as dSprites and Car3D. It also achieves significantly higher performance on the real-world dataset CelebA.

## 1 INTRODUCTION

Learning a disentangled representation is a long-desired goal in the deep learning community (Bengio et al., 2013; Peters et al., 2017; Goodfellow et al., 2016; Bengio et al., 2007; Schmidhuber, 1992; Lake et al., 2017; Tschannen et al., 2018). A disentangled representation matches how humans understand the world and provides many other benefits besides model interpretability (Bengio et al., 2013; Chen et al., 2016; Kulkarni et al., 2015). And it usually needs much fewer labels to learn challenging downstream tasks (van Steenkiste et al., 2019). It also generalizes much better even in face of examples generated by an unseen combination of the attribute values (Achille et al., 2018).

Given its importance and potential large impacts on downstream applications, disentangled representation learning has recently attracted great attention. Previous research has proposed a lot of methods, either built on variational auto-encoders (Kingma & Welling, 2013), such as $\beta$-VAE (Higgins et al., 2016) and FactorVAE (Kim & Mnih, 2018), or generative adversarial network (Goodfellow et al., 2014), such as InfoGAN (Chen et al., 2016) and InfoGAN-CR (Lin et al., 2020). Those methods have achieved preliminary successes on synthetic datasets such as dSprites (Matthey et al., 2017) and 3Dshapes (Burgess & Kim, 2018).

Albeit the initial successes, those synthetic datasets are limited in many aspects, for example, the background is usually clean and composed of a single color, the number of objects is small and the objects are mainly 2D without texture and occlusions. It is still an open question to scale those models to real-world complex datasets. It is even non-trivial to scale basic generative models to complex datasets to learn disentangled features, such as ImageNet (Deng et al., 2009).

In this paper, instead of studying disentanglement feature learning with a generative model, we investigate whether the discriminatively-trained contrastive models have the disentanglement property. Contrastive learning is a class of self-supervised learning methods that pull two augmentations of the same image close. The recent contrastive methods have achieved state-of-the-art performance on

the image pretraining tasks. Representative methods include MoCo (He et al., 2020), BYOL (Grill et al., 2020), SimCLR (Chen et al., 2020) and SwAV (Caron et al., 2020), etc. Contrastive learning has been proven to learn good visual representations from large-scale datasets. We continue to investigate the disentanglement property of their learned representations in this work.

To our surprise, we find that the widely used BYOL algorithm without any auxiliary loss exhibits strong feature disentanglement property. However, the disentanglement of contrastive learning is a weaker form of disentanglement. It follows the pattern that a representation dimension is disentangled to correspond to a single factor but a single ground truth factor might appear in multiple latent feature dimensions. We name this type of disentanglement as "group disentanglement". Although group disentanglement is a weaker form of disentanglement, we hypothesize that directly learning a compact and disentangled representation might be hard, due to the lottery ticket hypothesis (Frankle & Carbin, 2018). Pursuing group disentanglement instead of full disentanglement might be necessary to achieve disentanglement on the complex real-world dataset. The reason is probably that real-world images usually contain more details and noise, and a "factor" might be always correlated to some other visual existence due to dataset bias. A good example is a ship that usually comes together with a large area of blue background in existing datasets.

Further, we find that contrastively trained representation achieves the state-of-the-art FactorVAE disentanglement score when evaluated on established benchmarks, such as Car3D (Reed et al., 2015), dSprites (Matthey et al., 2017) and SmallNORB (LeCun et al., 2004). Besides the widely used synthetic benchmarks, we also evaluate the contrastive method on the CelebA (Liu et al., 2018) human faces real-world dataset. We find that it also achieves better or comparable performances than the other methods on five commonly used disentanglement metrics.

In summary, our contributions in this paper is mainly empirical and include the follows:

1. We show that a contrastive method, in particular BYOL, learns "group-disentangled" representations, without any extra auxiliary losses.

2. The contrastive method achieves the state-of-the-art performance on several widely used disentanglement learning benchmarks.

3. We propose to quantitatively evaluate disentanglements on a real-world dataset, which avoids the biases of synthetic images. Our contrastive method also achieves state-of-the-art performance on this benchmark.

## 2 RELATED WORKS

**Disentangled Representation Learning** Disentangled representation is desired as it represents a human interpretable pattern (Bengio et al., 2013; Chen et al., 2016; Kulkarni et al., 2015), enabling the downstream tasks learned more easily (van Steenkiste et al., 2019) and generalizes better (Achille et al., 2018). In this paper, we consider the fully unsupervised disentangled representation learning setting, i.e. we assume no annotations on which factors should be learned.

We notice the recent study of disentanglement is promoted by two communities: Disentanglement in Deep Features and Independent Component Analysis. Their research previously lie on different assumptions, data patterns, and evaluation metrics.

One community is motivated by the newly raised deep learning for encouraging disentangled representation over independent factors. they have shown much empirical progress on this problem and they directly term their goal as "disentanglement". The related study is usually based on deep generative models. For instance, VAE-based methods have achieved successes on this task (Higgins et al., 2016; Kim & Mnih, 2018; Chen et al., 2018; Kumar et al., 2017). Besides, Generative Adversarial Networks (GAN) (Goodfellow et al., 2014; Chen et al., 2016) are also put into the discussion of encouraging representations' disentanglement. More recently, people have shown that the GAN-based approach can achieve competitive performance as the above VAE variants (Lin et al., 2020; Jeon et al., 2018; Lee et al., 2020b). A recent work (Locatello et al., 2019) summarizes the popular methods and metrics in this community and proposes a tool for evaluation called *disentanglement_lib*, including popular metrics such as DCI (Eastwood & Williams, 2018), SAP (Kumar et al., 2017), MIG (Chen et al., 2018) and so on.

Besides this series of studies, exploring underlying factors of variation in data pattern is a long-standing goal of the Independent Component Analysis (ICA) community (Hyvärinen & Oja, 2000). They share many similarities, for example, generative models, e.g., VAEs, are recently popular in both (Khemakhem et al., 2020a; Klindt et al., 2020). ICA usually has different assumptions with the "purely unsupervised learning" (Hälvä et al., 2021). For example, the pattern of noise (Hyvarinen & Morioka, 2016; Khemakhem et al., 2020a) or some additional auxiliary variables (Hyvarinen et al., 2019; Khemakhem et al., 2020b) can be observed. Traditionally, ICA uses identifiability to assess their desired representation pattern and the popular metric is Mean Correlation Coefficient (MCC). SlowVAE (Klindt et al., 2020) recently makes a great effort to connect the two branches of study but it still requires additional information such as temporal transition pattern. As our study is for purely unsupervised learning and some assumptions of ICA methods can not be well matched, we mainly follow the settings and benchmark by *disentanglement_lib*k (Locatello et al., 2019).

**Contrastive Learning** Contrastive learning methods such as SimCLR (Chen et al., 2020), MoCo (He et al., 2020) and BYOL (Grill et al., 2020) have achieved great successes to learn good visual representation from no label. They create "views" by applying augmentations over images. They treat two views of the same image as "positive pairs", and views of all the other images as negatives. This setup is also known as examplar classification (Dosovitskiy et al., 2014), or instance discrimination (Wu et al., 2018). Representation learned by contrastive learning has shown great transfer capabilities to downstream tasks, such as object detection and semantic segmentation.

More recently, there are a lot of works trying to understand contrastive learning either theoretically (Wang & Isola, 2020; Arora et al., 2019; Tsai et al., 2020; Tosh et al., 2021; Tian et al., 2020b; Lee et al., 2020a) or empirically (Tian et al., 2020a; Zhao et al., 2020; Purushwalkam & Gupta, 2020). Zimmermann et al. (2021) suggests that the contrastive method can invert the data generation process. The conclusion is based on the analysis of Wang & Isola (2020) where negative samples are necessary and expected to be infinite. Zimmermann et al. (2021) make a good bridge between contrastive learning and independent analysis and study the model identifiability quantitatively in terms of MCC score. Compared with that, we focus more on more direct disentanglement analysis. Our contribution is more empirical but suggests the good disentanglement property of contrastive learning even without negative samples.

## 3 METHOD

In this paper, we explore whether contrastive methods learn a disentangled feature representation. If yes, under what condition it learns a disentangled representation. There are quite a few contrastive learning algorithms proposed (Grill et al., 2020; He et al., 2020; Chen et al., 2020; Caron et al., 2020). Although they differ on some specific aspects, they all aim to pull two augmentations of one image close. Without the uniformity property provided by negative samples (Wang & Isola, 2020), we find BYOL (Grill et al., 2020) still achieves unexpected good disentangled feature representation.

### 3.1 BYOL METHOD

BYOL is an unsupervised learning method that pulls two augmentation views of the same image close (Grill et al., 2020) to learn a high-level image representation. A significant difference of it against other contrastive learning is the absence of negative pairs during training. As shown in Figure 1, for each image $x$, we obtain two views of it: $x_1$ and $x_2$ by data augmentations. One of them goes through the online network stream, and the other goes through the target network stream. The target network's parameter is not trained by the gradient descent algorithm but set as the exponential moving average (ema) of the online network. Both the online and the target stream have a representation network (encoder) and a projection network. The online network has an extra prediction network after the projection network. The online stream's output $z_1$ and the target stream's output $z_2$ are pulled close to each other by requiring the two vectors to have similar directions in the latent space. More specifically, the loss function is

$$\mathcal{L} = -\frac{\langle z_1, z_2 \rangle}{\|z_1\|_2 \|z_2\|_2}.$$

In practice, the online representation network (without the projection or the prediction network) is usually the representation model for the downstream tasks. In this work, we follow the same convention, i.e. we study the disentanglement property of the output of the representation network.

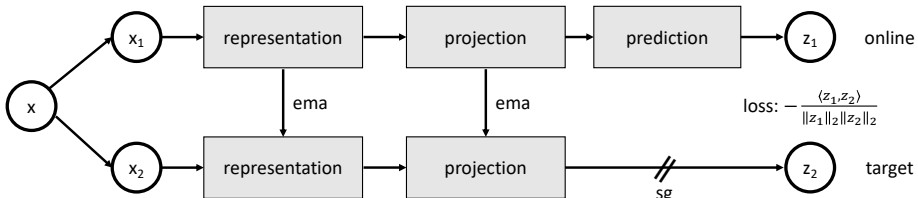

Figure 1: The BYOL network architecture. It has an online stream and a target stream. The online stream has a representation-projection-prediction pipeline while the target stream lacks the prediction part. The target branch is updated by exponential moving average of the online branch. The model is trained by encouraging the output of two streams from a positive pair of views to get close.

### 3.2 IMPLEMENTATION DETAILS

We follow most of the implementation details in the original BYOL paper, but we find that several details can be changed to achieve better feature disentanglement. For full implementation details please refer to the appendix A.1.

**Architecture** To make a fair comparison, we follow the encoder architecture used in Factor-VAE (Kim & Mnih, 2018) for all synthetic datasets, except that the latent code dimension is set to be 1000, since the contrastive learning requires a large latent code dimension to work (Grill et al., 2020). On CelebA, encoders are unified to be ResNet-50 (He et al., 2016).

**Normalization** By default, BYOL uses batch normalization throughout the network. However, we find that batch norm in encoder can reduce the feature disentanglement in some cases (see Section 5.5 for ablations). Thus we replace all batch norm with group normalization (Wu & He, 2018). Unless otherwise stated, the group number is set to 4 in our implementation by default.

**Augmentation** Augmentation is a crucial component for unsupervised learning methods as it provides a feasible way to create positive pairs in contrastive learning. In this work, we follow the default data augmentation used in BYOL, i.e., the composition of color jitter, graying, horizontal flip, gaussian blur, and random resize crop. We recognize that the parameter of random resize crop size is critical to learning disentangled features, which we will discuss in the ablation study.

## 4 MAJOR RESULTS

In this section, we summarize our major empirical findings. Section 5 will then continue to present experimental evidence to support these claims.

### 4.1 THE GROUP DISENTANGLEMENT PROPERTY

We find that the contrastive feature representation exhibits some form of disentanglement property: different ground truth factors map to different sets of latent dimensions, but multiple latent dimensions are corresponding to a single ground truth factor. We name this type of disentanglement as "group disentanglement". See Section 5.2 for experimental evidence to support this characteristic.

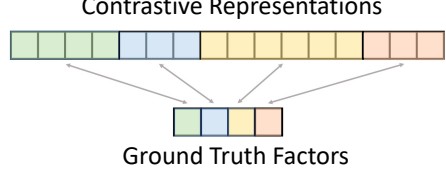

Figure 2: Illustration of group disentanglement. The lower vector denotes four ground truth factors and the upper vector is the group disentangled feature vector. Each ground truth factor may correspond to multiple feature dimensions, but each feature dimension only correspond to one factor.

While there is not a single widely accepted definition of the original disentanglement concept, we describe it intuitively as (1) **Disentanglement**: one dimension in the representation vector only represents one ground truth factor (2) **Compactness**: one ground truth factor only corresponds to

one dimension in the representation (3) **Completeness**: no ground truth factor is left out in the representation. Group disentanglement relaxes the compactness requirement, but only requires the disentanglement and the completeness. Figure 2 shows how group disentanglement looks like.

Group disentanglement is a weaker form of disentanglement, but we hypothesis that it might be a better proxy of disentanglement to scale to complex and realistic datasets. Directly learning the ideal compact disentanglement representation with neural network might be very hard because of the lottery ticket hypothesis (Frankle & Carbin, 2018). The lottery ticket hypothesis states that usually a well-trained neural network can be pruned to a much smaller network while keeping the same accuracy. However, it is much harder to directly train the same small network from scratch when randomly initialized. The hypothesis encourages a large randomly initialized network from start to contain a subset of the network that is the "winning ticket". We suspect that a compact disentangled representation is also hard to obtain for the same reason. Instead, group disentanglement might be a much more realistic goal to achieve.

### 4.2 Competitive Performance Across Benchmarks and Metrics

To quantitatively evaluate the disentanglement properties of the contrastive methods, we evaluate with existing benchmarks (Cars3D (Reed et al., 2015), dSprites (Matthey et al., 2017), Small-NORB (LeCun et al., 2004), Shapes3D (Burgess & Kim, 2018)) with various metrics. We find that although contrastive learning is not designed to learn disentangled feature representation, it still achieves better or comparable disentangled features when compared to existing specialized methods. We even build benchmarks on the real-world CelebA dataset for the first time, and the performance of contrastive learning is still robust to be top. See Section 5.3 for experimental support.

### 4.3 Batch Norm Discourages Feature Disentanglement

We find that batch norm in encoder consistently decreases the feature disentanglement level compared to no normalization layer (Section 5.5). By keeping the batch norm in the projector and the predictor, removing batch norm in the encoder will not cause model collapsing, which agrees with the observation in previous works (Richemond et al., 2020). On the contrary, replacing batch norm in encoder with group norm or layer norm will increase the feature disentanglement while achieving similar accuracy in downstream factor prediction. We notice that a similar phenomenon has been discovered before in supervised feature disentanglement. For example, Bau et al. (2017) discovered that a network trained with batch normalization layers has less interpretable (disentangled) neurons. We still do not fully understand this behavior, but we hypothesize that it may be caused by the shared batch statistics that make it hard for a feature to be aligned to the ground truth factor.

## 5 Experiments

In this section, we show the quantitative results to support the described observations above. We first introduce the experiment setup. Then we provide qualitative analysis about the learned representation pattern by contrastive learning. At last, we list a series of quantitative benchmark experiments to prove the good disentanglement property of BYOL on both synthetic and real-world datasets. Besides the main experiments in this section, more ablation experiments are provided in the appendix.

In this section, we aim to empirically understand the disentanglement properties of the contrastive method following the four questions. (1) What does the contrastive method learn in its latent representation? Is it disentangled? Can we visualize the latent space? (2) Quantitatively, how disentangled is the representation of contrastive methods? (3) How about real-world datasets? (4) What set of hyper-parameters best promotes the learned representation to be disentangled?

### 5.1 Experiments Setup

In this section, we describe the experimental setup, including the datasets we evaluate, the metrics for a quantitative study, and the previous disentangled feature learning algorithms for comparison.

**Datasets** Previous works evaluate representation disentanglement only on synthetic datasets, such as dSprites, Cars3D, SmallNORB, and Shape3D. Besides those datasets, we also include a real-world dataset CelebA. CelebA contains human celebrity faces images with 40 binary attributes annotations. The attributes include fine-grained properties of the human face such as whether wearing glasses or whether has wavy hair. Table 6 in Appendix explains the dataset details.

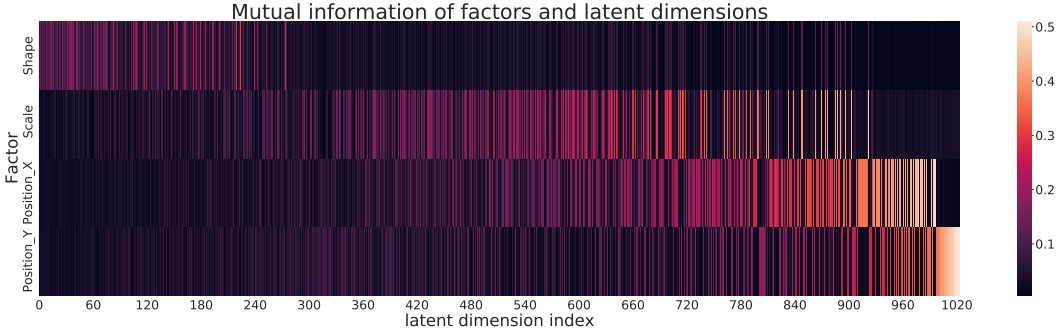

Figure 3: The mutual information heatmap between factors and latent code dimensions. It explains the pattern of "group disentanglement" that a factor shares high mutual information with multiple dimensions of representation instead of only one dimension described by the perfectly compact disentanglement. On the other hand, a certain representation dimension will not have high mutual information with more than one factor.

**Metrics** We follow `disentanglement_lib` (Locatello et al., 2019) to use five popular metrics for evaluation, i.e., BetaVAE, FactorVAE, MIG, SAP, and DCI. We leave out the Modularity metric since Locatello et al. (2019) suggests that it is inconsistent with other metrics. Please refer to Appendix A.4 for details including the calculation of mutual information.

**Reference Methods** Most of the previous state-of-art disentangled feature representation learning methods are either built on VAE (Kingma & Welling, 2013) or GAN (Goodfellow et al., 2014). $\beta$-VAE (Higgins et al., 2016) introduces a hyper-parameter to adjust the KL constraint in VAE. FactorVAE (Kim & Mnih, 2018) and $\beta$-TCVAE use adversarial training to reduce the correlations on different dimensions of the latent code. DIP-VAE (Kumar et al., 2017) pull the posterior to a factorized prior. Besides the VAE-based methods, we also investi-

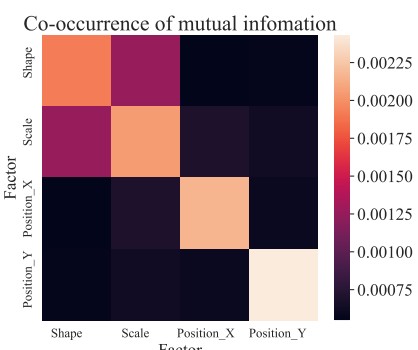

Figure 4: The visualization of normalized co-occurrence of mutual information on dSprites.

gate Generative Adversarial Networks (GANs). InfoGAN (Chen et al., 2016) encourages the latent code to have high mutual information with the generated image. IB-GAN (Jeon et al., 2018) and InforGAN-CR (Lin et al., 2020) add more constraints on InfoGAN to further promote disentanglement.

## 5.2 Understanding the Learnt Representation

In this part, we make the qualitative study of the disentanglement of representation learned by BYOL. Since contrastive methods are not generative models, it is hard to directly do factor-controlled pixel-wise reconstruction for visualization. We thus turn to measure the mutual information between learned features and ground truth factors to study that. The model is analyzed on dSprites dataset. dSprites has in total five factors (shape, scale, orientation, position_x, and position_y). But as explained in Appendix A.3, the orientation is ill-defined with ambiguity. For example, it is impossible to distinguish if a square rotates 0 degrees or 180 degrees. Therefore, we ignore this factor temporarily in the following discussion.

After encoding an input image to a representation vector, we compute the mutual information between each factor and each representation dimension. The mutual information between all dimensions and all four left factors are included in Figure 3. It shows an evident pattern that a ground truth factor can correspond to multiple representation dimensions but a single dimension of representation only has the high response to one factor.

To have a more intuitive understanding of to what extent a representation dimension may respond to more than one factor, we define metrics by the normalized co-occurrence of mutual information.

Given the mutual information between the representation vector and the $i^{th}$ factor, noted as $M_i$, $M_i^j$ is the mutual information by the $j^{th}$ dimension. Then, the normalized co-occurrence of mutual information between the $i^{th}$ factor and the $k^{th}$ factor is

$$\widehat{C}_{i,k} = \frac{\langle M_i, M_k \rangle}{||M_i||_2 \cdot ||M_k||_2} = \frac{\sum_{l=0}^{L} M_i^l M_k^l}{||M_i||_2 \cdot ||M_k||_2}.$$

where $L$ is the representation vector length. The definition conforms to the InfoMax (Linsker, 1988) principle of maximizing the mutual information of conceptually correlated pairs. We visualized the normalized co-occurrence of mutual information among the four factors by the learned representation in Figure 4. It agrees with the pattern revealed in Figure 3 that representation code on a certain dimension will not have high mutual information to more than one factor. Moreover, it shows the independence degree of factor pairs. For example, the shape and scale of dSprites objects are not fully disentangled because objects with the same scale value but in different shapes have different pixel area. The co-occurrence of mutual information on them is thus slightly higher than other pairs.

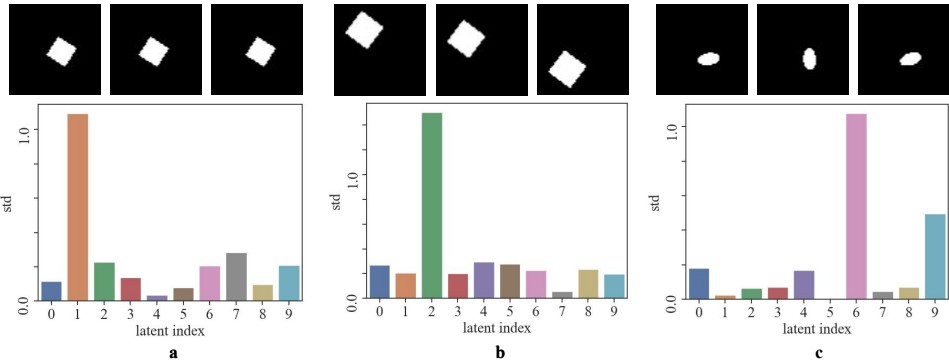

Figure 5: Representation variation when manipulating one factor only in the dimension-reduced version. In (a) and (b), *position_x* and *position_y* are manipulated respectively and only cause one dimension severely variate. While when manipulating the ill-defined factor *orientation*, two dimensions variate. The result shows a more compact disentanglement after reduction of dimension for those well-defined independent factors.

The analysis above recognizes the pattern of "group disentanglement". Then it comes a natural question of whether we can get a more compact and interpretable representation. Here, we reduce the representation dimension by PCA to 10. Figure 5 shows the result of latent variation when changing only one factor at once. Given three images with only one factor's value different, we generate the 10-dim representation vectors from them. Then, we compute the variance across the three vectors. The Figure 5(a) and (b) show how reduced latent code changes when manipulating *position_x* and *position_y* factor respectively. It shows good disentanglement that only one representation dimension has high variation. However, in Figure 5(c) we show a failure mode of the ill-posed factor *orientation* that change of factor causes both 6th and 9th dimension of reduced representation to have large variations. From the results, we observe that manipulating one well-defined independent factor causes evident variance on only one dimension. Compared with the original group disentanglement pattern, the dimension-reduced version becomes more compact and interpretable.

To summarize, we have found strong empirical evidence about the "group disentanglement" property of the representation learned by contrastive learning. Moreover, we have shown that the representation can be made more compact through proper unsupervised reduction.

## 5.3 COMPARISON ON EXISTING BENCHMARKS

To quantitatively evaluate the disentanglement of contrastive learning, we compare contrastive learning with the previous line of disentangled representation learning methods on commonly used benchmarks in this section. In the previous works, a lot of datasets have been proposed to evaluate the disentanglement learning method. Locatello et al. (2019) suggests that new method should be evaluated on a wide range of datasets to minimize potential dataset biases.

In this part, we follow the evaluation protocol of Locatello et al. (2019) to gain quantitative disentanglement measurement on dSprites dataset. For results on SmallNORB, Cars3D and Shapes3D, please refer to the results in Appendix. B. The results of VAEs are reported based on the best configuration from Locatello et al. (2019). We calculate the mean performance and standard deviation from the released raw logs. The results of InfoGAN, IB-GAN and InfoGAN-CR are reported in Lin et al. (2020) and we use them its officially released checkpoint to evaluate under our protocol. The performance of Ada-GVAE and Ada-ML-VAE are reported in a previous paper (Locatello et al., 2020). In the independent component analysis community, we can also find some methods to compare with but usually they are designed for different benchmarks and metrics. We adopt the method reported in ICE-BeeM (Khemakhem et al., 2020b) and migrate it to the benchmarks we evaluate on. However, as we focus on unsupervised learning setting, no factor attribute is available for training data, so the conditional denoising score matching (CDSM) version of ICE-BeeM can not be used here. We turn to use its unconditional version, and we term it as EBM (energy-based model) in following content.

| | Model | BetaVAE | FactorVAE | MIG | SAP | DCI |
|---|---|---|---|---|---|---|
| **VAE** | $\beta$-VAE | 82.3 (7.6) | 65.8 (9.2) | 26.3 (11.0) | 5.2 (2.7) | 39.3 (13.2) |
| | $\beta$-TCVAE | 86.7 (2.4) | 76.6 (7.8) | 23.8 (6.8) | 6.9 (0.9) | 36.3 (7.1) |
| | FactorVAE | 84.9 (2.8) | 75.3 (7.4) | 18.4 (9.0) | 6.8 (0.8) | 28.8 (10.6) |
| | DIP-VAE-I | 82.7 (3.3) | 59.1 (4.8) | 9.6 (5.1) | 5.2 (2.6) | 14.4 (4.6) |
| | DIP-VAE-II | 81.5 (4.9) | 58.6 (7.6) | 7.4 (3.4) | 3.6 (2.2) | 12.3 (5.2) |
| | AnnealedVAE | 86.5 (0.1) | 60.1 (0.0) | **35.2 (1.3)** | 7.6 (0.5) | 37.9 (2.1) |
| | Ada-GVAE | 88.0 (2.7) | 73.1 (3.9) | 17.3 (4.7) | 6.6 (2.0) | 32.3 (4.6) |
| | SlowVAE | 87.0 (5.1) | 75.2 (11.1) | 28.3 (11.5) | 4.4 (2.0) | 47.7 (8.5) |
| **ICA** | EBM | 82.3 (2.0) | 65.7 (12.5) | 1.7 (0.5) | 3.0 (1.2) | 19.1 (1.8) |
| **GAN** | InfoGAN | – | 59.0 (7.0) | – | – | – |
| | IB-GAN | – | 80.0 (7.0) | – | – | – |
| | InfoGAN-CR | 85.5 (1.0) | 88.0 (1.0) | 19.8 (3.2) | 6.0 (1.0) | 14.0 (5.2) |
| **CL** | BYOL (Ours) | **93.2 (0.4)** | **91.6 (0.8)** | 29.3 (0.4) | **8.0 (0.4)** | **66.9 (0.2)** |

Table 1: Mean and standard deviation (s.d.) metric scores on dSprites dataset. The results of BYOL are averaged over three random seeds. The results of Ada-GVAE and SlowVAE refer to the Slow-VAE paper (Klindt et al., 2020). It shows strong and robust disentanglement property of BYOL.

All evaluation settings follow the configuration of Locatello et al. (2019). The results strongly suggest that contrastive learning can well promote disentangled features. Moreover, it shows the evidence that different metrics can not be precisely aligned as they use different perspectives to measure the disentanglement degree, which itself evens lacks a unified definition. Moreover, as the latent dimension of VAE-based methods is only 128, smaller than that of BYOL (1000-d), we increase the representation dimension of VAEs to 1000 for a more fair comparison. The results are in Table 11 in Appendix, showing that increasing latent dimension does not boost the VAEs' disentanglement. Therefore, the performance gap between BYOL and VAE-based methods does not come from the change of representation dimension.

## 5.4 COMPARISON ON REAL-WORLD DATASETS

Although the four datasets above can establish how well each method performs, those datasets are all synthetic and are limited in various aspects. For example, dSprites only has one object per image on a black background; SmallNORB contains synthetic images of 3D objects viewing from different angles, but only contains 50 instances within 5 categories. High performance on those datasets does not necessarily transfer to real world images, where there might be a lot more variations and unseen objects during test time. To get a clearer sense of how each of the methods would perform on the real world datasets, we further evaluate our method and several other methods on CelebA.

For evaluation of MIG and SAP metrics, to make a fair comparison, the representation vector of all methods is reduced to 40 dimensions. The results are shown in Table 2. BYOL continues to show high disentanglement compared with other methods.

| | Model | BetaVAE | FactorVAE | MIG | SAP | DCI |
|---|---|---|---|---|---|---|
| **VAE** | VAE | 21.5 | 6.1 | 0.8 | 0.9 | 11.2 |
| | $\beta$-VAE | 19.1 | 5.8 | 0.1 | 0.6 | 8.7 |
| | $\beta$-TCVAE | 19.9 | 9.8 | 0.6 | 1.2 | 3.5 |
| | FactorVAE | 25.3 | **12.0** | 0.4 | 0.6 | 7.1 |
| | DIP-VAE-I | 21.0 | 9.3 | 0.2 | 0.9 | 13.8 |
| **GAN** | InfoGAN-CR | 16.8 | 11.3 | 1.6 | 2.8 | 22.0 |
| **CL** | BYOL (Ours) | **35.7** | 11.5 | **2.6** | **8.2** | **41.0** |

Table 2: Disentanglement evaluation on the CelebA dataset. The result shows great robustness of BYOL's learned representations to show disentanglement on real-world datasets. Yet, the large gap between the score from that on synthetic datasets emphasises the difficulty of learning disentangled factors on real-world images.

| **normalization** | w/o norm | BN | GN | LN | IN |
|---|---|---|---|---|---|
| **FactorVAE score** | 89.6 | 85.0 | **91.6** | 91.4 | 19.1 |

Table 3: Results of using different normalization strategies on dSprites. For group normalization, we set group number to 4. BYOL collapses with instance normalization (IN). For complete results, refer to Table 13 and Table 14 in the appendix.

## 5.5 Effects of Hyper-Params

Hyper-parameters significantly affect the performance of contrastive learning. In this section, we systematically study the influence on disentanglement property from some recognized crucial inductive bias for contrastive learning.

The batch normalization in BYOL has been considered the reason why BYOL does not collapse even without negative pairs for long. However, a previous work (Richemond et al., 2020) finds BYOL can avoid collapsing even without batch statistics. But the normalization strategy between layers is still recognized as a key variant of BYOL. Thus we experiment with five normalization layers configuration in the encoder network on the dSprites dataset. The results are shown in Table 3. We find the commonly used BN decreases the disentanglement performances. No normalization already does well regarding feature disentanglement. Group norm (Wu & He, 2018) and layer norm (Ba et al., 2016) are the two best normalization techniques. On dSprites, instance norm (Ulyanov et al., 2017) completely breaks the contrastive learning process. We provide more study on normalization in Table 13 and Table 14 in Appendix where more variables, such as batch size, are taken into consideration. From all the studies, it remains unclear why batch norm could harm the feature disentanglement. However, this phenomenon was also empirically observed in the supervised learning literature as well. For example, in the Bau et al. (2017) paper, the author shows that CNN with batch norm layer has significantly fewer interpretable concepts than those without BN.

We also perform an ablation study on other hyper-parameters. For example, we find the representation trained on one dataset can show transferrable disentanglement to other datasets and BN causes model collapse with a big learning rate (Table 12). More details are provided in Appendix D.

## 6 Conclusion

In this paper, we show some empirical study on the disentanglement property by contrastive learning. We find with minor modification, the existing BYOL framework can achieve state-of-the-art disentanglement performance on multiple benchmarks and under diverse quantitative metrics. Besides, the qualitative study reveals the different "disentanglement" nature of contrastive learning that the representation actually shows "group disentanglement". It is thus a relaxed form of disentanglement due to weaker bound of compactness. To the best of our knowledge, we are the first to build quantitative disentanglement benchmark with contrastive learning involved and the first to build benchmark of disentanglement measurement on real-world datasets (CelebA). Recently, the study of contratsive learning, or generally self-supervised learning, is still motivated by empirical observations. We wish our work can reveal some clues on this line of study.

# 7 REPRODUCIBILITY

In this section, we provide the information required to reproduce our results reported in the main text. And we commit to making the code implementation and evaluating checkpoints public.

**BYOL implementation**    For the implementation details of BYOL, please refer to Appendix A.1. The model architecture, training setups, and dataset preprocessing are all explained in detail.

**VAE methods implementation**    For evaluation on synthetic datasets, i.e., dSprites, Cars3D, SmallNORB and Shapes3D, the disentanglement score is from the original logs of `disentanglement_lib` Locatello et al. (2019) [1]. In the released logs, each method has different training configurations, and our reported result is from the configuration with the highest average performance overall provided random seeds. For evaluation on CelebA dataset, we follow an open-sourced implementation in Pytorch [2] and align the encoder architecture of all methods to be the same as described in Appendix A.1. Parameters are kept as the default well-tuned version in the provided implementation.

**GAN methods implementation**    It is hard to insert GAN methods' performance in the benchmark as the training is not always stable and the discriminator weights are usually not provided in many public codebases. When evaluating on synthetic datasets, the FactorVAE scores of InforGAN, IB-GAN, and InfoGAN-CR are provided in the paper of Lin et al. (2020). But the evaluation of other metrics in Lin et al. (2020) uses a not aligned settings with Locatello et al. (2019), so we check its officially release [3] to reevaluate the provided implementation and model weights under the unified evaluation setup. We perform the same evaluation process for results on CelebA dataset.

**Energy-based Model (EBM)**    We refer to the implementation of ICE-BeeM (Khemakhem et al., 2020b) for this method. We use the officially released codebase for it [4]. The encoder implementation has been aligned with our default already. The only modification we make is to use the unconditional version instead of its default conditional version in loss computation. Please refer to the **runners/real_data_runner.py** file for details. Besides, we add the dataset definition for dSprites, SmallNORB, Cars3D, and Shapes3D to it as the implementation for our method. All hyperparameters are kept the default in its released version.

**Evaluation Protocol**    Though our study is focused on disentanglement property, the evaluation of a representation model is two-step. In the first step, we perform factor prediction as a downstream task based on the learned representation model. Only when the accuracy of factor prediction is high enough to ensure the model weights "valid", we would go to the evaluation of disentanglement metric scores. For the implementation details of factor prediction, please refer to Appendix A.3. For disentanglement metrics evaluation, we use the official implementation of Locatello et al. (2019), i.e. `disentanglement_lib`. The settings of some important parameters in Appendix A.4.

---

[1] https://github.com/google-research/disentanglement_lib

[2] https://github.com/AntixK/PyTorch-VAE

[3] https://github.com/fjxmlzn/InfoGAN-CRhttps://github.com/fjxmlzn/InfoGAN-CR

[4] https://github.com/ilkhem/icebeem

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

# A APPENDIX

## A.1 IMPLEMENTATION OF BYOL MODEL

**Architecture** We describe the implementation details of BYOL in this part. To make a fair comparison with previous methods, for synthetic datasets, i.e. dSprites, Cars3D, SmallNORB, and Shapes3D, we follow the encoder architecture in Factor VAE (Kim & Mnih, 2018). The pipeline details are shown in Table 4. After each shown convolutional layer in the figure, there is a ReLU activation layer. By default, there is also a group normalization (group number = 4) layer. So, the encoder is a stack of (Conv-ReLU-GN) blocks. For CelebA dataset, the encoder is the commonly adopted ResNet-50 (He et al., 2016) backbone. By default, the final output channel number is 1000, i.e, $D = 1000$. We note here that the group normalization we use as default has nothing necessarily connected with the group disentanglement property we study.

Besides the representation network (encoder), BYOL also has a projector network and a predictor network. Both of them consists of a pipeline "Linear $\longrightarrow$ BN $\longrightarrow$ ReLU $\longrightarrow$ Linear". The projection dimension is 256 and the hidden dimension of the projector is 4096. The predictor keeps a 256-dimensional feature vector in its pipeline.

| Encoder of BYOL |
| --- |
| **input**: $64 \times 64$ images |
| **pipeline**: |
| 4×4 conv, stride 2, 32-channel |
| 4×4 conv, stride 2, 32-channel |
| 4×4 conv, stride 2, 64-channel |
| 4×4 conv, stride 2, 64-channel |
| 4×4 conv, stride 2, 128-channel |
| 1×1 conv, stride 1, $D$-channel |

Table 4: The encoder architecture for our implemented BYOL on synthetic datasets. By default, we set $D = 1000$ to be aligned with the commonly used ResNet-50 backbone network. Besides, there is a ReLU activation layer and a possible normalization layer following each convolutional layer to create a stack of (Conv-ReLU-Norm) blocks.

**Training settings** We make minor modification on the training setting of BYOL. For training on all datasets, the images are resized to 64x64. For the data preprocessing, we copy 1-channel images of dSprites and SmallNORB to 3-channel. During training stage, we use such a pipeline of augmentation (in *PyTorch*-style):

1. *RandomApply(transforms.ColorJitter(0.8, 0.8, 0.8, 0.2), p=0.3)*
2. *transforms.RandomHorizontalFlip()*
3. *RandomApply(transforms.GaussianBlur((3,3), (1.0, 2.0)), p=0.2)*
4. *transforms.RandomResizeCrop(size=(64, 64), scale=(0.6,1.0))*
5. normalization

For dSprites and SmallNORB, image pixel value is uniformly normalized from [0,255] to [0,1.0]. For Cars3D, Shapes3D, and CelebA, we adopt the commonly used Imagenet-statistic normalization for preprocessing the image values.

During training, we use Adam optimizer by default, whose learning rate is $3e-4$ without weight decay. The batch size is set to be 512 without exceptional notation. For evaluation on dSprites, Shapes3D and CelebA, we select the weights after training for 15 epochs for evaluation. We select the weights after training for 140 epochs for evaluation on Cars3D and weights of the 200th epoch on SmallNORB considering the small scale of these two datasets.

As explained in some ablation experiments, the default training setting is not the one we could get the highest disentanglement score. For example, using Group Normalization + Weight Standardization achieves a higher disentanglement score than only using Group Normalization or not using normalization. But we aim to make minor modifications to the original BYOL implementation and follow previously commonly used setups.

To decrease the influence of randomness, we train each model configuration multiple times with different random seeds. We report the results by the average and three random seeds. To be precise, as our implementation is based on Pytorch, we would set initialize the libraries of *numpy*, *torch*, *torch.cuda* and *random* with the same random seeds. We use 0, 1, 2 as the three values of random seeds in trails.

## A.2 IMPLEMENTATION OF VAE MODELS ON CELEBA

The benchmark (Locatello et al., 2019) provides VAE-based methods' disentanglement performance on synthetics datasets only, e.g., dSprites, Shapes3D, Cars3D, and SmallNORB. For the real-world CelebA dataset, we need to train the models by ourselves to gain the results in Table 2. We use an open-sourced implementation [5] for VanillaVAE, $\beta$-VAE, Factor-VAE, $\beta$-TCVAE and DIP-VAE-I. We follow the provided well-tuned hyper-parameters provided by them. Please refer to the repository for details.

| **Linear classifier for factor prediction** |
|---|
| **input**: $D$-dim latent code |
| **pipeline**: |
|       linear layer $(1000 \longrightarrow 256)$ |
|       ReLU |
|       linear layer $(256 \longrightarrow 128)$ |
|       ReLU |
|       linear layer $(128 \longrightarrow 100)$ |
|       ReLU |
|       $K$ linear layers (in parallel, $100 \longrightarrow n_k, 1 \le k \le K$ ) |

Table 5: The implementation of linear classifier for factor prediction. With the $D$-dim latent code from the encoder, the classifier has multiple fully connected layers to shrink the feature vector to 100-dimensional. Then, given the number of factor types $K$ for the target dataset, as each factor has $n_k (1 \le k \le K)$ values, we have $K$ linear layers following the last shared layer in parallel. These layers predict the factor value on the corresponding $K$ factors.

## A.3 IMPLEMENTATION OF FACTOR PREDICTION

As mentioned in the main paper, for each gained representation model, i.e., encoder, we will first confirm it "valid" by training a linear classifier for factor prediction. In the stage, the encoder weights will be frozen and only the attached classifier trained. For the task on all datasets, we use a unified classifier as shown in Table 5. The classifier has stacked fully connected layers (linear layers) and multiple in-parallel heads for factor prediction. On different target datasets, the number of factors and the possible values of factors vary. For the datasets we evaluate on, the value range and the accuracy threshold to confirm the quality of a representation acceptable are shown in Table 6. We use a well-trained BetaVAE model weights to do the factor prediction first. Its prediction accuracy serves as the threshold in this sanity check stage. Usually, the accuracy threshold is set to be 0.8 or higher. But for the orientation factor on dSprites and Shapes3D, we relax the constraint to 0.5. The reason is that the factors are ill-defined to have ambiguity. For example, when a square or ellipse object in dSprites has an orientation of 180 degrees or 0 degrees, the image can be the same while the factors of orientation are regarded differently. For this reason, these factors bring noise into

---

[5]https://github.com/AntixK/PyTorch-VAE/

our study and validate the model weights. We, therefore, relax the accuracy requirement for them in practice. Besides, SmallNORB is the most difficult dataset by our observation. The factor prediction accuracy on it is usually much lower, especially for azimuth and elevation, both of which are even harder to be predicted.

| Dataset | Factor | Value Range | Accuracy Threshold |
|---------|--------|-------------|---------------------|
| **dSprites** | Shape | 3 (square, ellipse, heart) | 0.95 |
| | Scale | 6 | 0.95 |
| | Orientation | 40 | 0.50 |
| | Position X | 32 | 0.80 |
| | Position Y | 32 | 0.80 |
| **Shapes3D** | Floor hue | 10 | 0.80 |
| | Wall hue | 10 | 0.80 |
| | Object hue | 10 | 0.80 |
| | Scale | 8 | 0.80 |
| | Shape | 4 | 0.80 |
| | Orientation | 15 | 0.50 |
| **Cars3D** | Elevation | 4 | 0.80 |
| | Azimuth | 24 | 0.80 |
| | Object id | 183 | 0.80 |
| **SmallNORB** | Instacne category | 10 | 0.70 |
| | Elevation | 9 | 0.40 |
| | Azimuth | 18 | 0.40 |
| | Lighting condition | 6 | 0.90 |
| **CelebA** | 40 human face factors | binary | 0.80 |

Table 6: The factor of datasets we evaluate on. Some factors are originally continuous but discretized into all integers. Therefore, all factor prediction is classification task. Given a representation model trained on the training set of a dataset, the linear classifier should achieve accuracy higher than the theshold on all factors to be recognized "valid". Only a "valid" representation model would be put into the next step for disentanglement score evaluation. For the details of these factor definition, please refer to the original papers of these datasets.

## A.4 EVALUATION METRICS

Five metrics are used to quantify the disentanglement of our model.

**Beta Vae Metrics** Introduced in Higgins et al. (2016), beta vae metrics assumes each dimension corresponds to one category in linear classifier. Representations are obtained after generated samples with only one factor k fixed. Calculating the summation of the divergence between different representations and putting this result into a linear classifier, we train a model that possibly outputs the corresponding k. The accuracy of this linear model is the value of beta vae metrics.

**Factor Vae Metrics** Kim & Mnih (2018) argues the beta vae metrics has the tendency to fail into a spurious disentanglement, and proposes a new metrics based on a linear classifier. Representations are obtained after generated samples with only factor k fixed. Normalizing each dimension in representations in terms of standard deviation. Index of dimension with lowest variances of normalized representation and the factor index k is the input/output for the linear classifier. The accuracy of this linear model is the value of factor vae metrics.

**Mutual Information Gap** Chen et al. (2018) assumes the disentanglement model has the property that most information of one specific factor is contained in one dimension or a group of certain dimensions. The mutual information gap is the summation of the difference of highest and second-highest normalized mutual information between fixed k factor and dimensions in representations as shown below:

$$\frac{1}{K} \sum_{k=1}^{K} \frac{1}{H_{z_k}} (I(v_{j_k}, z_k) - \max_{j \neq j_k} I(v_j, z_k)) \tag{1}$$

Where K is the overall number of ground truth factors, $z_k$ is the factors of latent variables and $j_k = \arg\max_j I(v_j, z_k)$.

**DCI disentanglement** As Eastwood & Williams (2018) suggests, disentanglement is the entropy of the relative importance, evaluated with a Lasso or a Random Forest classifier, of each dimension for specific factors. Completeness is the entropy of the possibility that one factor is captured by a single code variable. Informativeness, overlapping with Disentanglement, is the prediction error of the model predicting factors.

**SAP** Kumar et al. (2017) proposes the Separated Attribute Predictability (SAP) score. A score metrics is computed with classification score of predicting $j^{th}$ factors on $i^{th}$ dimension as the $ij^{th}$ entry. SAP is the mean of the difference of the highest and second-highest scores for each column.

For evaluation protocol, we follow the implementation provided by Locatello et al. (2019). Despite exception, the evaluation batch size is 64, the *prune_dims.threshold* is 0.06. If a classifier is required to be trained during evaluation, *num_train* is 10000 and *num_eval* is 5000. For Mutual information computation, the used discretizer function is the histogram discretizer and the number of bins in discretization is 20. For evaluation of MIG and SAP on dSprites, SmallNORB, Cars3D, and Shapes3D, BYOL representation vectors are reduced to 10 dimensions by PCA to be aligned with other methods. For evaluation of MIG and SAP on CelebA, to have a fair comparison, the representation vectors of all methods are reduced to 40 dimensions.

|     | Model | BetaVAE | FactorVAE | MIG | SAP | DCI |
|-----|-------|---------|-----------|-----|-----|-----|
|     | $\beta$-VAE | **100.0 (0.0)** | 89.3 (1.2) | 11.7 (1.1) | 1.4 (0.9) | 38.7 (4.6) |
|     | $\beta$-TCVAE | **100.0 (0.0)** | 92.2 (2.7) | **15.5 (2.9)** | 1.7 (0.3) | 42.7 (3.5) |
| **VAE** | FactorVAE | **100.0 (0.0)** | 91.7 (4.1) | 10.6 (2.2) | **2.0 (0.5)** | 29.0 (6.7) |
|     | DIP-VAE-I | **100.0 (0.0)** | 90.5 (5.0) | 5.9 (2.8) | 1.9 (1.4) | 22.6 (5.6) |
|     | DIP-VAE-II | **100.0 (0.0)** | 85.0 (6.1) | 5.1 (2.7) | 1.3 (0.8) | 20.8 (5.4) |
|     | AnnealedVAE | **100.0 (0.0)** | 85.0 (4.3) | 7.6 (1.0) | 1.5 (0.5) | 18.5 (4.3) |
|     | SlowVAE | **100.0 (0.0)** | 90.4 (0.5) | 15.4 (2.2) | 1.6 (0.5) | 48.0 (2.4) |
| **CL** | BYOL (Ours) | **100.0 (0.0)** | **95.8 (1.2)** | 7.6 (0.9) | 1.8 (0.7) | **48.5 (2.3)** |

Table 7: Evaluation of disentanglement on Cars3D by different metrics. The results of BYOL are averaged over three random seeds. It shows strong and robust disentanglement property of BYOL.

# B MORE BENCHMARK RESULTS

We report the benchmark results on dSprites in the main text (Table 1). We continue to report the full benchmark results on other datasets in the Locatello et al. (2019) evaluation protocol, i.e., Cars3D, SmallNORB, and Shapes3D.

As the large-scale benchmark of Locatello et al. (2019) provides the original logs on Cars3D, and SmallNORB datasets, we simply report the best configuration VAE-based methods trained by them. The original logs on Shapes3D are not available, so we train and evaluate on Shapes3D by ourselves and the output results are well aligned with the reported result in Locatello et al. (2019). The results of other methods come from the same resources as those on the dSprites benchmark. As we find the energy-based model (EMB) implemented from ICE-BeeM (Khemakhem et al., 2020b) fails to

learn valid representation on Cars3D with its default configuration and hyperparameter settings, we do not report it here. By default, we report mean scores averaged on three random seeds for our method. But on Shapes3D, as the mean scores for some methods are not available, we report the median performance instead as the default in Locatello et al. (2019). The evaluation protocol is the same as that for dSprites as described in Appexdix A.4.

The full results are shown in Table 7, Table 8 and Table 9 respectively. On SmallNORB and Cars3D datasets, our method still achieves the state-of-the-art performance in terms of BetaVAE score, FactorVAE score, and DCI. But its MIG and SAP results are outperformed by other methods. This is because the BYOL approach learns a group-disentangled representation. Therefore, there are multiple dimensions in the representation denoting the same ground truth factor. MIG and SAP score computes the mutual information difference between the most related and second related dimension to each ground truth factor. Thus a group disentangled feature can have a low MIG and SAP score, even after the dimension reduction. On the other hand, on Shapes3D, our method fails to make the state-of-the-art performance on all metrics and the gap between its performance and others on MIG is even larger. Given the good performance of other methods on Shapes3D, we need more knowledge about our method's relative failure on this benchmark.

|  | Model | BetaVAE | FactorVAE | MIG | SAP | DCI |
|---|---|---|---|---|---|---|
| **VAE** | $\beta$-VAE | 84.1 (2.7) | 60.1 (2.4) | 25.0 (1.1) | 11.4 (1.1) | 32.6 (0.6) |
|  | $\beta$-TCVAE | 84.5 (2.7) | 60.3 (2.3) | 25.4 (0.9) | 11.7 (1.1) | 35.2 (0.7) |
|  | FactorVAE | 80.8 (3.8) | 62.5 (3.6) | 23.9 (2.0) | 10.2 (0.9) | 33.4 (1.1) |
|  | DIP-VAE-I | 84.2 (3.2) | 69.8 (4.6) | 24.3 (2.7) | 10.2 (1.4) | 30.0 (2.1) |
|  | DIP-VAE-II | 85.2 (1.3) | 58.4 (2.1) | **25.5 (1.5)** | **14.4 (0.4)** | 32.3 (0.7) |
|  | AnnealedVAE | 60.8 (6.2) | 50.0 (9.9) | 9.1 (2.2) | 6.8 (0.8) | 15.7 (6.4) |
|  | SlowVAE | 78.2 (3.8) | 47.0 (2.9) | 23.8 (1.8) | 7.8 (1.1) | 28.7 (0.7) |
| **ICA** | EBM | 79.0 (4.4) | 57.9 (3.5) | 1.7 (0.5) | 1.9 (0.1) | 13.9 (2.2) |
| **CL** | BYOL (Ours) | **97.0 (0.8)** | **81.0 (0.5)** | 3.3 (0.9) | 2.2 (0.3) | **51.0 (1.0)** |

Table 8: Evaluation of disentanglement on SmallNORB by different metrics. The results of BYOL are averaged over three random seeds. It shows strong and robust disentanglement property of BYOL.

To conclude, on the benchmarks and under the evaluation protocol provided by Locatello et al. (2019), our implemented contrastive learning shows SoTA-level performance in most cases. Its downsides on MIG and SAP metrics are probably related to its relaxed version of disentanglement property. The standard benchmark results help to provide a more accurate sense of the disentangling representation ability of contrastive learning.

## C More Qualitative Study

Limited by the main text length limitation, we make a more qualitative study about the disentanglement property shown by the contrastive learning here. In Figure 4, we show the co-occurrence of mutual information of factors on dSprites by our model. We perform some qualitative studies on SmallNORB, Cars3D, and Shapes3D as well. The visualization results are shown in Figure 6, Figure 8 and Figure 10. We can have some observation from them.

**SmallNORB** For the results on SmallNORB in Figure 6, though most non-diagonal entries have very low co-occurrence of mutual information, two pairs of factors show slightly higher co-occurrence. They are "azimuth-elevation" and "object/instance category-lighting". After investigating the dataset, we find the two pairs of factors are not fully independent. Figure 7 show some samples with corresponding factors manipulated. We could see that the elevation and azimuth are not fully independent. And the relation between the instance/object category and the lighting factor is even more obvious because the lighting condition is sensible by the shadow on and under the

|      | Model       | BetaVAE      | FactorVAE   | MIG         | SAP        | DCI         |
|------|-------------|--------------|-------------|-------------|------------|-------------|
| **VAE** | $\beta$-VAE    | 98.6         | 83.9        | 22.0        | 6.2        | 58.8        |
|      | $\beta$-TCVAE  | 99.8         | 86.8        | 27.1        | 7.9        | 70.9        |
|      | FactorVAE   | 94.2         | 82.5        | 27.0        | 6.1        | 67.2        |
|      | DIP-VAE-I   | 95.6         | 79.7        | 15.2        | 4.0        | 55.9        |
|      | DIP-VAE-II  | 97.8         | 88.4        | 18.1        | 6.3        | 41.9        |
|      | AnneledVAE  | 86.1         | 80.9        | 35.9        | 6.2        | 47.4        |
|      | Ada-ML-VAE  | **100.0**    | **100.0**   | 50.9        | 12.7       | 94.0        |
|      | Ada-GVAE    | **100.0**    | **100.0**   | 56.2        | **15.3**   | **94.6**    |
|      | SlowVAE     | **100.0 (0.1)** | 97.3 (4.0) | **64.4 (8.4)** | 5.8 (0.9) | 82.6 (4.4) |
| **ICA** | EBM        | 75.9 (11.2)  | 53.2 (8.7)  | 5.2 (2.2)   | 2.8 (1.1)  | 21.8 (11.0) |
| **CL** | BYOL (Ours) | 91.5 (3.9)   | 82.5 (2.4)  | 5.2 (1.7)   | 2.8 (0.3)  | 53.1 (1.5)  |

Table 9: Evaluation of disentanglement on Shapes3D by different metrics. The results of BYOL are averaged over three random seeds. It shows strong and robust disentanglement property of BYOL. Because the original experiment logs on Shapes3D by *disentanglement_lib* is not released, we can not get the average performance of baseline models. Instead we report median disentanglement scores in this table by referring to results reported in Locatello et al. (2020) but the std error is not available. The median performance of SlowVAE is reported in its original paper (Klindt et al., 2020).

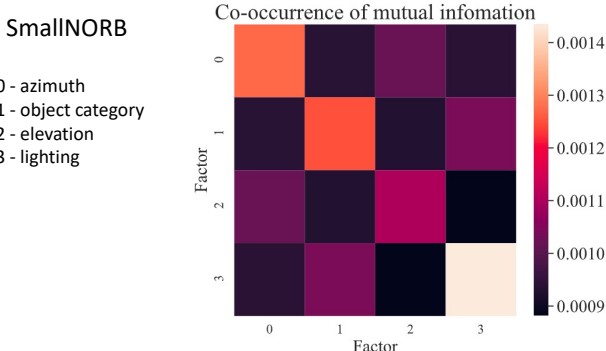

Figure 6: The visualization of co-occurrence of mutual information of the factors of SmallNORB.

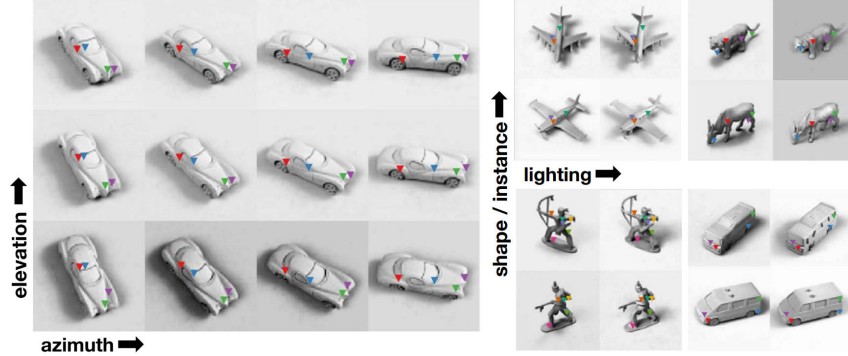

Figure 7: Samples from SmallNORB dataset. The variance is controlled by the factor indicated on axis. The image is from Jakab et al. (2018).

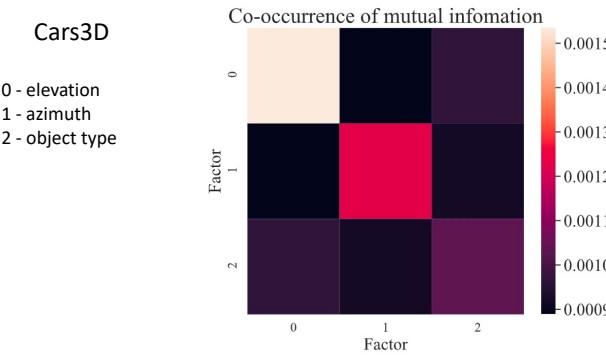

Figure 8: The visualization of co-occurrence of mutual information of the factors of Cars3D.

instance, whose distribution and shape is highly determined by the instance category. This reveals a common observation that, in real-world data (SmallNORB is not real-world yet but it is complicated than 2D synthetic images), the factors are usually hard to define to be fully independent or disentangled.

**Cars3D** For the results from Cars3D shown in Figure 8, only one pair of factors show some co-occurrence, i.e. "elevation-object type". To study how this happen, we randomly sampled some data from Cars3D by different object types and elevations as shown in Figure 9. It shows that with the same value of elevation, samples of different object types help different visual elevation. So these two factors are not well disentangled. This might explain the slightly higher co-occurrence of mutual information between this pair of factors.

**Shapes3D** The results on Shapes3D are shown in Figure 10. The result shows relatively bad disentanglement. To be precise, some factor pairs show low mutual information co-occurrence as expected, such as the color factors of floor, wall, and object and the pair of "object color - azimuth" . But the MI co-occurrence of "wall color - object size" and "object color - object size;" are higher than we expected as we did not recognize their high dependence. This result might relate to our model's relatively poor performance on Shapes3D as well.

We hope the additional results shown above are helpful to provide more sense of the dataset configuration and what we should expect from a well-disentangled representation model. The full benchmark results show that contrastive learning can achieve SoTA-level performance on standard datasets under some metrics while the performance is not that good for SAP/MIG in some cases. A study of this observation is important as it shows the disagreement of existing disentanglement metrics. Besides, as the latent dimension of our implemented BYOL model is relatively high and the group disentanglement property relaxes the requirement of compactness in disentanglement, making dimension-wise disentanglement not distinguishable as expected in compact disentanglement. Because SAP and MIG are measured by comparing dimension-wise mutual information differences, the property of group disentanglement is the main potential reason for the observed disagreement of disentanglement metrics from the reported benchmarking results.

## D    MORE ABLATION EXPERIMENTS

Limited by the main content page length, we put some additional ablation study here to help better understand the influence of important inductive bias when studying representation disentanglement.

In the original implementation of contrastive learning, the min-ratio of Random-Resize-Crop is usually very small, e.g, 0.08 for BYOL or 0.2 for MoCo/SimCLR. However, we find this hyper-parameter is critical. We thus perform a study with different minimum ratios of random-resize-crop. The results are shown in Table 10. From the table, we find the disentanglement score first increases when increasing the min-ratio of random-resize-crop but then drops. The default min-ratio is set to be 0.6 in our implementation.

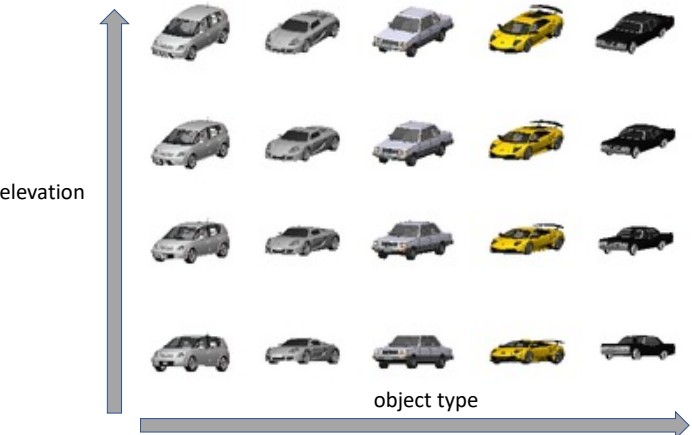

Figure 9: Some samples from the Cars3D dataset. The factor of azimuth is the same for all samples. The factors of samples vary along the axis of object type and the axis of elevation. We could find these two factors are actually not fully independent.

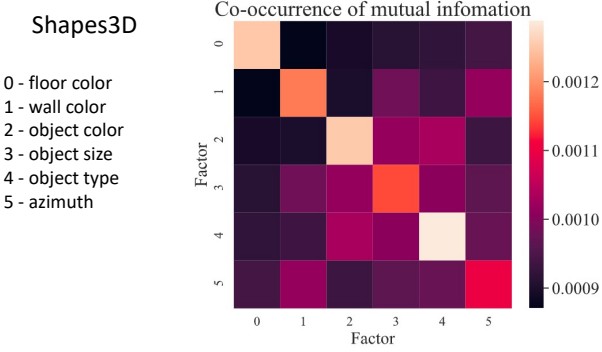

Figure 10: The visualization of co-occurrence of mutual information of the factors of Shapes3D.

| min-scale | BetaVAE | FactorVAE |
|:---:|:---:|:---:|
| 0.1 | 93.4 | 86.6 |
| 0.2 | 95.4 | 87.0 |
| 0.3 | 94.4 | 84.8 |
| 0.4 | **96.0** | 86.8 |
| 0.5 | 94.8 | 88.3 |
| 0.6 | 93.8 | 88.0 |
| 0.7 | 93.5 | **89.0** |
| 0.8 | 91.6 | 81.4 |
| 0.9 | 87.7 | 77.8 |

Table 10: The influence of minimum scale of random resize-crop augmentation on the disentanglement of BYOL learned features. Here, the batch size is set to be 64.

| method | BetaVAE | FactorVAE | MIG | SAP |
|--------|---------|-----------|-----|-----|
| VAE | 79.6 | 50.0 | 4.3 | 0.3 |
| $\beta$-VAE | 20.3 | 24.5 | **7.4** | 0.2 |
| $\beta$-TCVAE | 62.1 | 54.7 | 1.1 | 0.3 |
| Factor-VAE | 63.2 | 54.0 | 0.2 | **0.8** |
| DIP-VAE-I | **82.0** | **67.3** | 0.7 | 0.2 |

Table 11: Disentanglement performance of VAE-based methods on dSprites, with an increased dimension of latent code to 1000-dimensional.

Learning rate is also another important hyper-parameter, we use Adam optimizer with learning rate to be $3e - 4$ as default. But we are still interested in the influence of learning rate to the disentanglement property of learned representation. At the same time, batch normalization in the encoder network is another critical component. In this part, we aim to study the transferability of learned representation, so we train BYOL under with different learning rates and batch normalization on three datasets respectively. But the evaluation is always performed on the test set of dSprites. The results are included in Table 12. It shows, no norm not just shows a higher disentanglement score than BN but is also more robust with the variance of learning rate. When the learning rate is high, e.g., 3e-3 or 3e-2, the model with BN has collapsed already while the model without norm can still output a reasonably high disentanglement score. On the other hand, on the transferability from one dataset to another, the disentanglement score also shows high agreement.

| | dSprites | | 3dShapes | | CelebA | |
|---|---|---|---|---|---|---|
| **learning rate** | w/ BN | w/o BN | w/ BN | w/o BN | w/ BN | w/o BN |
| 3e-5 | 75.6 | 83.0 | 60.5 | 77.0 | 59.6 | 78.3 |
| 6e-4 | 83.2 | 92.0 | 64.3 | 72.4 | 67.3 | 81.1 |
| 1e-3 | 80.9 | 91.8 | 65.2 | 72.3 | 63.7 | 81.0 |
| 3e-3 | 42.3 | 91.4 | 23.1 | 71.0 | 65.1 | 80.5 |
| 3e-2 | 19.1 | 85.9 | 17.0 | 70.6 | 25.9 | 80.3 |

Table 12: We study the impact of learning rates over trained model's disentanglement. As all models are all tested on dSprites benchmark, BYOL shows its good generalization ability when trained on one dataset but transferring to other datasets.

As the supplement of study on the normalization strategy, we provide a more detailed study in Table 13 where five normalization strategies are evaluated with different batch sizes, which was found crucial in choosing normalization strategy in visual tasks. The result of the table still suggests that Batch Normalization weakens BYOL's disentanglement property and no normalization, group normalization, and layer normalization show good disentanglement with different batch sizes for training. When using instance normalization in the representation network, BYOL goes to collapse on dSprites.

Richemond et al. (2020) shows that it is possible to achieve similar results when replacing BN with group norm and weight standardization (Qiao et al., 2019). We continue to show the influence of involving Group Normalization (GN) and weight standardization (WS) in Table 14. Here, we keep setting the group number to 4 and batch size to 512. And we use the two weight decay values in the original BYOL and Richemond et al. (2020) respectively (3e-8 and 1.5e-6). The results show that compared with solely using GN, adding additional Weight Standardization can not just help learn features of higher quality but also be more disentangled.

| batch size | w/o norm | BN | GN | LN | IN |
|:---:|:---:|:---:|:---:|:---:|:---:|
| 64 | 88.5 | 80.0 | **90.4** | 89.2 | 19.1 |
| 128 | 89.6 | 83.3 | **91.6** | 91.0 | 19.1 |
| 256 | 92.0 | 85.6 | 93.2 | **93.8** | 19.1 |
| 512 | 89.6 | 85.0 | **91.6** | 91.4 | 19.1 |
| 1024 | 86.1 | 83.1 | **88.0** | 85.9 | 19.1 |
| 2048 | **84.5** | 79.8 | 82.5 | 82.8 | 19.1 |
| 4096 | **82.4** | 75.8 | 82.2 | 81.9 | 19.1 |

Table 13: Results of using different normalization strategy with different batch size during training on dSprites. We evaluate the FactorVAE score to indicate the disentanglement property of model weights from dSprites. For group normalization, we set group number to 4. BYOL collapses with instance normalization (IN) only and the evaluated disentanglement score from it also collapses to a constant here.

| normalization | BetaVAE | FactorVAE | MIG | SAP |
|:---|:---:|:---:|:---:|:---:|
| w/o norm | 92.9 | 89.6 | 16.4 | 5.6 |
| GN | 93.2 | 91.6 | 29.4 | **8.0** |
| GN + WS | 96.6 | 91.6 | 31.5 | 7.4 |
| GN + WS (wd = 3e-8) | **96.7** | **93.5** | 31.3 | 7.6 |
| GN + WS (wd = 1.5e-6) | 96.2 | 19.1 | **32.2** | 7.5 |

Table 14: The results to study the influence of Group Normalization (GN) and Weight Standardization (WS) on the representation disentanglement. The results prove the effectiveness of both GN and WS to help promote representation disentanglement.

