# OpenReview forum: "Disentangling Properties of Contrastive Methods"
_ICLR.cc/2022/Conference — ICLR 2022 Submitted_

### Official Review · Reviewer_xhwR · 2021-10-31

**Correctness:** 2
**Technical Novelty And Significance:** 2
**Empirical Novelty And Significance:** 3
**Recommendation:** 3
**Confidence:** 4

**Main Review:**

Before knowing that we were assigned this paper for review, we were excited about the claims of this paper and replicated the results. We are therefore now familiar with the paper. Although the results in the paper can be replicated, the claims are a little misleading. We will explain below:

Strengths:
* The observation that BYOL features have better disentanglement properties than specialized methods is interesting.
* The observation that group disentanglement appears for the dSprites and Shape3D datasets is interesting, but its importance might be overstated
* A useful study of the effect of the normalization method on the disentanglement quality.
* The paper for the most part is nicely written and has attractive figures

Weaknesses:
* The presented metrics are incomplete. DCI is not provided on most of the synthetic experiments. Beyond dSprites (which is an easy dataset) only factorVAE is reported which gives a limited comparison between the methods.
* Misleading claim - representations are group disentangled - first of all, we did not find this to be correct for SmallNORB which is the hardest synthetic dataset. More importantly, we found that even for the easiest dataset, the only reason why it looks to experience group disentanglement is that mutual information is only computed at the single dimension level. However, if we take the 20 dimensions that are least imformative about an attribute, we could still classify the attribute with near perfect accuracy. So the claim of group disentanglement is misleading, there are just a group of dimensions that each individually seems to have information about a distinct factor, but the group is not disentangled at all.
* I'm not sure if this is actually claimed but it might be confusing to the reader who might think that group normalization is related to group disentanglement . In fact, the experiments showed this is not the case. Choosing a single group is fine. So it's really about replacing batchnorm by layernorm type normalization.
* No analysis is provided to explain why BYOL representations have these attributes. As the method here is not applicative by itself, its main utility is the theoretical understanding. We not think that such an understanding is provided here.

**Summary Of The Paper:**

This paper makes an interesting empirical observation - BYOL representations have better disentanglement properties, according to some metrics, than current specialized methods. Furthermore, the authors found that the selection of the normalization function affects the results. The authors also claim the dimensions are "group disentangled" although this is only shown on one dataset.

**Summary Of The Review:**

To summarize, the empirical observation that BYOL features are better disentangled than other methods is interesting. However, there are two important limitations: i) there is very little analysis trying to explain why this happens, and as no applications were shown, better understanding and analysis should be the main product of this research but they are lacking. ii) the experiments are incomplete - the group disentanglement is only shown for one dataset, our replication did not observe it for SmallNORB, such selection of results might make readers reach the wrong conclusions about the properties of the method. I suggest more complete experiments and more analysis.

###################

Our discussion with the authors and the poor experimental design reduced our confidence in this paper - we change our recommendation to rejection.

---

> ### Author Response · Authors · 2021-11-17
> **Response to Reviewer xhwR**
>
> Dear reviewer,
>
> Thank you very much for confirming the strength of this paper and proposing valuable comments to it for improvement. For the responses to your questions, please refer to the following.
>
> **Q1.  The presented results are incomplete.**
>
> A1. In the original version, we report only FactorVAE scores for some benchmarks, the revised version has more complete results. We notice that MIG and SAP scores are not as good as other metrics. We believe this is because the originally high dimension of BYOL makes MIG and SAP hard to be high, which measures the difference of the highest MI dimension and the second highest MI dimension. In the group disentanglement, these two dimensions probably fall into the same "group" contributing the same factor.
>
> **Q2. Misleading claim about "group disentanglement"**
>
> A2. We tried to reproduce your findings from our trained checkpoint but failed. To be precise, we try to use 20 dimensions randomly selected from a 1024-dim representation to predict factors on SmallNORB. To eliminate variables, we use the same classifier architecture and training hyperparameters. The 20-dim representation is upsampled to 1024-dim by copy-paste to fit the classifier configuration. The details are stated below.
>
> (1) **On SmallNORB.** we found a drop of factor accuracy on all factors of SmallNORB by 5% - 15% if using 20-dim representations. And the accuracy from both the original representation and the 20-dim representation are both far away from perfect. As the result from 20  lowest informative dimensions should serve as a lower-bound, our observation here looks different from that of yours.
>
> (2) **On easier dataset.** We try on dSprites. We tried to randomly select the 20-dim out of the 1024-dim representation for 20 times. And most subsets show significant performance drop. Because dSprites does not provide official splitting, we use our own train-test splitting here (train:test=2:1). We provide the code, dSprites splitting file and checkpoints to reproduce this experiment through [this anonymous link](https://drive.google.com/drive/folders/1n8gZZFWoKXiDDEQeXxo-7wbDaLtze_rp?usp=sharing) which can reproduce our results in one command. In case the reviewer used BYOL with BN by mistake, we also tried our BN version implementation and provide that in the link. The details of accuracy on dSprites are (shape, scale, orientation, position-x, position-y):
> * The full-representation accuracy of GN version is  **0.952, 0.874, 0.279, 0.861, 0.779**
> * A sampled 20-dim representation accuracy of GN version is **0.253, 0.199, 0.024, 0.033, 0.037**
> * The full-representation accuracy of BN version is  **0.984, 0.985, 0.502, 0.927, 0.914**
> * A sampled 20-dim representation accuracy of BN version is **0.211, 0.172, 0.027, 0.038, 0.059**
>
> We tried our best to reproduce your report. Please let me know if we have any divergence on the understanding of the paper or we made any fault in our reproduction.
>
> **Q3. Is there any relation between group normalization and group disentanglement?**
>
> A3. We didn't make a claim about the connection between group normalization and group disentanglement in the paper. We will make it clear that they are not related. We agree with you that layernorm also performs well, better than BN. In the appendix, we make the ablation study about it. In our implementation, we use GN with group=4 which provides a more convenient baseline for the later ablation study of GN+Weight Standardization which is paid attention to by the contrastive learning community.
>
> **Q4. No theoretical understanding is provided in this paper.**
>
> A4. Our contributions in this paper are purely empirical. However, we think these empirical findings themselves are interesting enough to many people. Actually, a lot of work in deep learning, including some of the most impactful ones, first came out without any theoretical guarantees, but they turn out to be very useful in practice later. Some of them were widely used for many years, before theoreticians gave rigorous theoretical analysis.
>
> Thank you,
>
> Authors

---

### Official Review · Reviewer_DZ9Z · 2021-11-02

**Correctness:** 4
**Technical Novelty And Significance:** 3
**Empirical Novelty And Significance:** 3
**Recommendation:** 8
**Confidence:** 2

**Details Of Ethics Concerns:**

I do not have ethics concerns for this work.

**Main Review:**

Strengths:
- The simplicity of the method: BYOL with group norm instead of batch norm in the encoder.
- The proposed concept of group-disentanglement is a more relaxed but also more realistic property in real world scenarios. My feeling is that redundancies are needed in both model parameters (as shown in lottery ticket and dropout) and representations (group disentanglement).
- There are comprehensive experiments and ablation studies to support the claims in the paper: on synthetic datasets, a real world dataset CelebA (which I think is a significant step to study disentanglement), and important hyperparameters.


**Summary Of The Paper:**

This work explores the disentangling properties of contrastive methods. The authors discover that contrastive methods, particularly BYOL, learns disentangled representations with just a change of normalization method in the encoder. The work also proposes a new concept called "group disentanglement", which is a relaxed version of the original disentanglement. BYOL learns representations with group disentanglement and achieves SOTA on disentanglement benchmarks on not only synthetic image datasets but also a real world dataset.

**Summary Of The Review:**

I am not very familiar with the field of disentanglement study, so I am not able to judge very well the significance of this work. However I like the simple and clear idea (to study disentanglement properties of contrastive methods) and the comprehensive experimental results. I especially like the concept of group disentanglement and the improvement of benchmarks on a real world dataset. Therefore I am recommending an accept.

---

> ### Author Response · Authors · 2021-11-17
> **Response to Reviewer DZ9Z**
>
> Dear reviewer,
>
> Thank you so much for the review feedback and positive comments. We added discussion in the general response for the motivation and contribution of this paper. Besides, we also have discussions with other reviewers. Please refer to that for more information on this paper. The unsupervised learning is highly motivated by empirical study at this moment and we continue to provide empirical findings on its disentanglement properties. And all these are done without negative samples. We believe we are the first to provide the observation and benchmark that onto the popular standard disentanglement research benchmark. We hope the findings can benefit the development of representation learning and provide support for  theoretical study.
>
> Thank you,
>
> Authors

---

### Official Review · Reviewer_K9hY · 2021-11-02

**Correctness:** 3
**Technical Novelty And Significance:** 1
**Empirical Novelty And Significance:** 2
**Recommendation:** 5
**Confidence:** 4

**Main Review:**

- Contributions: As your results are purely empirical I think statements like "We show that a contrastive method learns group-disentangled representations" might be troublesome since readers could expect theoretical guarantees.
- Contributions: You say that your contrastive method (which is confusing since you just the usual BYOL method without modifying it) achieves state-of-the-art performance; however, I do not see a comparison with reasonable models, e.g., [1], [2], [3].
- Abstract: The statement "Prior methods achieved initial successes on simplistic synthetic datasets but failed to scale to complex real-world datasets." is unclear - to which prior methods are you referring?
- Related work/Discussion: (1) Regarding the Zimmermann et al. 2021 paper, you say they didn't perform any quantitative analysis. However, this paper even proposes a new dataset of 3D objects and demonstrates the success of a contrastive method on it. (2) Furthermore, the comparison seems a bit skewed (please correct me if I'm missing something here), since your primary point of criticism of that work are the (strong) theoretical assumptions - yet, your paper does not contain any theoretical but just empirical results. (3) Finally, you say that you "build a series of quantitative benchmarks", yet, later in the paper, it seems like you only re-use existing datasets. If you mean that you tried evaluating disentanglement on real-world datasets, you should cite previous papers that did so, e.g. [2]. Can you please specify what you mean by "build" and make this more precise?
- Major Results: How does your group disentanglement property differ from existing metrics in this field? I think a discussion should be added here. Your current description of it via Disentlement, Compactness, and Completeness sounds a lot like the metric suggested by [4] (i.e., the DCI metric) - what makes it special?
- Major Results: I don't really see how the lottery ticket hypothesis is connected to disentanglement. At the heart of your argument, you say that you only say you suspect this to be the reason but do not give any reasoning as to why you think so - you only describe the lottery ticket hypothesis before.
- Table 1, 2: Over how many random seeds did you average? What is the error of the score? That information is necessary to decide whether the results of BYOL are significantly better than those of the other methods.
- Table 3, 4: Is it correct that these results correspond to just a single random seed?
- Table 4, 5.5: You only discuss the result of BN; however, it remains unclear whether the BN results are really significantly worse than the other results. Furthermore, there is no discussion of the IN results, even though, that number is drastically lower than that of BN.
- Introduction/Related work/Results: You did not include the best-working VAE-based disentanglement method, namely SlowVAE [1].
- 5.4: (1) How is the reduction to 40 feature dimensions performed? Do you mean that you changed the architecture such that the output is only 40 dimensional or did you post-process the 1000 and 128-dimensional features, respectively? (2) Given that CelebA does not contain ground-truth latent variables, how did you compute the disentanglement metrics? This is a crucial aspect that is missing at the moment.

[1]: Klindt, David, et al. "Towards nonlinear disentanglement in natural data with temporal sparse coding." \
[2]: Khemakhem, Ilyes, et al. "ICE-BeeM: Identifiable Conditional Energy-Based Deep Models Based on Nonlinear ICA." \
[3]: Zimmermann, Roland S., et al. "Contrastive Learning Inverts the Data Generating Process." \
[4]: Cian Eastwood and Christopher KI Williams. A framework for the quantitative evaluation of disentangled representations.

**Summary Of The Paper:**

The paper applies the contrastive BYOL method on a set of datasets, which are mostly well-established in the disentanglement community.  Here, they show empirical evidence that BYOL successfully learns disentangled representations.

**Summary Of The Review:**

Even though the idea of the paper to look further into the disentanglement properties of contrastive methods is interesting, I cannot recommend accepting it in its current form due to a lack of comparison with existing methods, impreciseness in the language/results, and too few explanations about the observed results.

**Update**: After the rebuttal, I increased my score by one level from 3 to 5 - yet, there are still open issues that prevent me from recommending acceptance.

---

> ### Author Response · Authors · 2021-11-17
> **Response to Reviewer K9hY (Part I)**
>
> Dear reviewer,
>
> Thank you so much for the detailed review and valuable suggestions. Your comments raise many downsides of this paper which we will be improving the paper based on. For responses to your questions, please refer to the following points.
>
> **Q1. Not enough theoretical guarantees for the statement of "group disentanglement"**
>
> A1. We are sorry that our statement might make readers expect theoretical guarantees. We will tune our writing to explicitly emphasize our contributions are purely on the empirical side.
>
> **Q2. Our implementation is just the usual BYOL without modifying it or not.**
>
> A2. Our implementation is based on BYOL but made some modification on its default configurations, for example we use group normalization to replace batch normalization as we find GN shows better disentanglement . Since the batch normalization has long been thought of as a key design in BYOL[R1], we feel this is a critical change. Besides this, some other modifications are made on data augmentation and encoder architecture.
>
> **Q3. Comparison with other reasonable models.**
>
> A3. We follow a popular benchmark provided by disentanglement_lib[L1] for evaluation. When choosing models to compare with, we noticed some methods from the ICA community but there was a practical difficulty that they work on different benchmarks and under different metrics (usually MCC). It is challenging to adapt them to our setups. As far as we know, SlowVAE[1] is the first work trying to merge the two genres in the disentanglement community and ICA community. Besides, a recent work[H1] analyses the difference and potential connection of these two communities. As we work on purely unsupervised learning settings, we select some reasonable methods to add into the new evaluation results (see pointers of tables in the general response). For the methods your mention:
>
> (1) SlowVAE[1] is a good practice and provides already benchmarking results, we should have added this. But as SlowVAE focused on obtaining disentangled representations by exploiting the temporal transition pattern which is not available for still image datasets so we could only report its results on the original datasets (UNI, in Table 1, 16, 17, 18, 19 of paper[1]).
>
> (2) ICE-BeeM[2] tried to learn representation by a conditional energy-based model. To compare with it, we migrate its officially released codebase (https://github.com/ilkhem/icebeem) to our settings but use the unconditional version to fit the unsupervised learning settings. We followed its default hyperparameter settings on real-world datasets but tuned the learning rate and training epoch and aligned the data augmentation.We tried our best to make its best performance but it still failed to gain reasonable disentanglement scores on Cars3D. The migration of codebase and evaluation script will be made public.
>
> (3) The paper[3] provides a connection between the contrastive learning and independent component analysis. They report results on different datasets under the MCC metric. The work and ours are both in the field of contrastive learning, we believe our work is a good extension to theirs, empirically, to benchmark contrastive  learning with the popular disentanglement evaluation protocols. Moreover, a key difference is that we show contrastive learning can achieve significant disentanglement even without negative samples.
>
> **Q4.  The statement "Prior methods achieved initial successes on simplistic synthetic datasets but failed to scale to complex real-world datasets." is unclear.**
>
> A4. The sentence is from the abstract where we can't put a citation. The extension is actually in the second and third paragraph in the introduction section. For "prior methods", we meant the VAE-based methods and some GAN methods, which are usually studied on synthetic datasets for disentanglement study. We will tune the writing to make the statements more accurate and well connected.
>
> References:
>
> [1]: Klindt, David, et al. "Towards nonlinear disentanglement in natural data with temporal sparse coding."
>
> [2]: Khemakhem, Ilyes, et al. "ICE-BeeM: Identifiable Conditional Energy-Based Deep Models Based on Nonlinear ICA."
>
> [3]: Zimmermann, Roland S., et al. "Contrastive Learning Inverts the Data Generating Process."
>
> [R1] Richemond, Pierre H., et al. "BYOL works even without batch statistics."
>
> [L1] Locatello, Francesco, et al. "Challenging common assumptions in the unsupervised learning of disentangled representations."
>
> [H1] Hälvä, Hermanni, et al. "Disentangling Identifiable Features from Noisy Data with Structured Nonlinear ICA."

---

> > ### Author Response · Authors · 2021-11-17
> > **Response to Reviewer K9hY (Part II)**
> >
> > **Q5. Questions regarding related works [3]**
> >
> > A5. We answer your questions by points:
> >
> > (1) **Quantitative analysis in [3]**: Our word was not properly used by saying they didn't perform quantitative analysis. We meant no quantitative results on the disentanglement metrics of the benchmark we follow are provided. They evaluated on the identifiability mnetrics MCC. We would make a more proper statement of this part.
> >
> > (2) **Criticism of theoretical results in [3]**: In the related work, our focus is that the work[3] has theoretical analysis but goes necessarily under the existence of negative samples in contrastive learning and the amount is expected to be large or even infinite. Our work instead studies disentanglement property into the case where no negative samples exist and onto the popular disentanglement evaluation protocols. Yes, our contribution is empirical. We do like the path provided by [W1] and the following [3] on theoretically analysing contrastive learning. But we realized the theoretical development in our case should fail onto a totally different path, which is what we are working on.
> >
> > (3) **Did we build quantitative benchmarks?**: We follow a popular disentanglement evaluation protocol and benchmarks on the real-world dataset CelebA for the first time. When we say we build a benchmark, we mean the use of CelebA dataset for evaluating the disentanglement.  Some ICA works such as [2] also make some quantitative evaluation on real-world dataset but under different metrics and properties of interest. Besides, the setup of [2] is not fully unsupervised, some auxiliary variables are used in the conditional version of ICE-BeeM. We will cite corresponding work that also evaluates on real-world datasets, and thank you for the pointers.
> >
> > **Q6. Difference between group disentanglement property of the existing metrics.**
> >
> > A6. We use **disentanglement, compactness and completeness** to describe the common expectation of disentanglement, which is, as you say, described as **disentanglement, completeness and informativeness** in [4]. While as explained in Section 4.1, just after mentioning the three aspects of disentanglement, group disentanglement is defined to *"relaxes the compactness requirement but only requires the disentanglement and completeness"*. This makes group disentanglement different from the common disentanglement, to be precise, a relaxed form.
> >
> > **Q7. How the lottery ticket hypothesis is connected to disentanglement.**
> >
> > A7. As we explained above, the difference between group disentanglement and normal disentanglement is that group disentanglement drops the compactness requirement. i.e. It means that the learned representation can have multiple dimensions representing the same concept. This is in contrast to the original disentanglement requirement, where it requires no two dimensions to represent the same ground truth factor.
> >
> > Our argument is “the compact disentangled features might be hard to learn due to the lottery ticket hypothesis”. We think this is true because the lottery ticket hypothesis says that even if there is a compact small network that can achieve high performance for a task, if you don’t have the correct random initializations for that small network, it would not be possible for the learning algorithm to find it. Our hypothesis is that directly learning the compact disentangled representation might be hard, because the network has a bottleneck in the final representation and it might not be possible for the optimization algorithm to find the optimal answer. This potential hardness suggests that a group disentanglement representation might be a good relaxation to the original compact disentanglement.
> >
> > **Q8. Random seed and error of scores.**
> >
> > A8. In the original version paper, all reported numbers come from averaging results over three random seeds. We will add the note in each evaluation part. Besides, we add the standard deviation of results in the new evaluation tables. Please refer to the general response above and the revised paper for details.
> >
> > References:
> >
> > [2]: Khemakhem, Ilyes, et al. "ICE-BeeM: Identifiable Conditional Energy-Based Deep Models Based on Nonlinear ICA."
> >
> > [3]: Zimmermann, Roland S., et al. "Contrastive Learning Inverts the Data Generating Process."
> >
> > [4]: Cian Eastwood and Christopher KI Williams. A framework for the quantitative evaluation of disentangled representations.
> >
> > [W1] Wang, Tongzhou, and Phillip Isola. "Understanding contrastive representation learning through alignment and uniformity on the hypersphere."

---

> > > ### Author Response · Authors · 2021-11-17
> > > **Response to Reviewer K9hY (Part III)**
> > >
> > > **Q9. Discussion about BN and IN.**
> > >
> > > A9. In the original paper, we report a simple study of normalization in Table 4 showing that GN (our default choice) is better than BN to get disentangled representations. In section 5.5 we could make a pointer to Table 11 and Table 12 for more complete ablation study and discussion. GN is better than BN in all batch size choices and IN fails to learn a useful representation (collapse). Moreover, inspired by a discussion of normalization in BYOL[R1], we also studied the effect and combining GN and weight standardization (WS). People are still trying to figure out why BYOL can avoid collapse without negative samples and the normalization has been paid much attention. In this work, we provide the study over it from the perspective of disentanglement.
> > >
> > > **Q10. Missing reference SlowVAE[1].**
> > >
> > > A10.  It was our negligence. We have added it into the new version and some discussion. Moreover, we also put into the evaluation as discussed above. But one thing about SlowVAE is that it focuses on using temporal transitions so in fact they build new datasets based on the standard still image datasets we use. We could only compare with its version without using transition pattern.
> > >
> > > **Q11. Dimension reduction to compute mutual information.**
> > >
> > > A11. For evaluating MI-related metrics on CelebA, we reduce the dimension to 40 to align with other methods and to 10 on other datasets. We use PCA to do that. We discussed it in Appendix A.4 of the original version of paper. We will add it in the main text before the evaluation.
> > >
> > > **Q12. How the attribute/factor is determined on CelebA.**
> > >
> > > A12.  CelebA is a dataset of human faces and 40 binary attributes are annotated as described in Section 5.1 in the original version of paper. We use them as 40 factors which is discussed in previous Table 7 (now Table 6) in the paper for details.
> > >
> > > Thank you,
> > >
> > > Authors
> > >
> > > References:
> > >
> > > [1]: Klindt, David, et al. "Towards nonlinear disentanglement in natural data with temporal sparse coding."
> > >
> > > [R1] Richemond, Pierre H., et al. "BYOL works even without batch statistics."

---

> > > > ### Comment · Reviewer_K9hY · 2021-11-29
> > > > **Response**
> > > >
> > > > Thank you for clarifying these points. Regarding Q8: Table 2 and 3 still do not show the standard deviation which makes it impossible to judge how substantial the difference between the methods (Table 2) and normalization function (Table 3) is - this should be added, too.
> > > > All in all, I think this revised version improved the quality of your submission! However, my main concern regarding the motivation/position of the paper remains: if your main message is that CL yields disentangled representations, then this is has been already shown by a previous study (see above); if it is that BYOL works like SimCLR in that regard, then (1) a comparison with the above method as well as (2) a theoretical explanation of the results would improve the relevance of this paper. Also, there are still a lot of typos/grammar mistakes in the text which should be corrected. Taken together, I can raise the score slightly but think that your paper is not yet ready for publication.

---

> > > > > ### Author Response · Authors · 2021-11-29
> > > > > **the main message we want to send in this work**
> > > > >
> > > > > We would like to emphasize that previous work[3] claims InfoNCE loss can invert the data generation process, which is not the same as the group disentanglement we are claiming here. [3] claims that the InfoNCE learnt representation is an orthogonal transformation of the ground truth factors. However, our empirical findings is a stronger version: each of the latent dimensions belong to only one ground truth factor. I.e. The transformation matrix is a block diagonal matrix. It is block diagonal since the latent dimensions are larger than the number of ground truth factors. In summary, our empirical findings are group disentanglement, while [3] claims to recover an orthogonal transformation of it.
> > > > >
> > > > > The second major difference is that we work on BYOL while [3] work on InfoNCE. Note that the proof in [3] is completely based on the InfoNCE loss and thus doesn’t imply our empirical findings here. The position of our paper is a general claim about group disentanglement properties in contrastive-style learning methods. And this more general property is also not covered by [3].
> > > > >
> > > > > We admit that our work lacks on the theory side, but we see great opportunities after presenting this work to the community. We are very excited to see future work that explains the phenomenon we find. More specifically, understand the following questions from a theoretical view: (1) A potentially unified theory that explains why two styles of CL methods both have the group disentangle property. (2) Why is the transformation matrix being a block diagonal, instead of a general orthogonal matrix? The current theory can not explain this, since the loss is the same after any rotation of the latent space.
> > > > >
> > > > > Our empirical work first points out this phenomenon. We feel like our work is the basis to introduce the community to those fundamental questions. We would like to thank you for your follow-up comments, which provide valuable revision for our improvement. We are improving the paper all the time, including polishing it. All grammar issues and typos would be checked.
> > > > >
> > > > > [3]: Zimmermann, Roland S., et al. "Contrastive Learning Inverts the Data Generating Process."

---

> > > > > ### Author Response · Authors · 2021-11-29
> > > > > **Standard deviation for Table 2 and Table 3**
> > > > >
> > > > > For the improved version of Table 2 and Table 3, please check the updated general response.

---

### Official Review · Reviewer_By54 · 2021-11-10

**Correctness:** 3
**Technical Novelty And Significance:** 4
**Empirical Novelty And Significance:** 2
**Recommendation:** 5
**Confidence:** 4

**Main Review:**

The paper makes the previously unreported finding that contrastive learning produces "group disentangled" representations and outperform other disentanglement models (as measured by standard disentanglement measures). It offers a nice literature review, exposition is clear and provides many implementation details that should help reproducibility.

Though presented results are solid, it would be interesting to see the complete set of results (probably in Appendix), namely Fig. 4 and Tab. 2 (or Tab. 1) for other datasets. It might greatly benefit from some insight (perhaps preliminary) on why contrastive representations are disentangled when supervised/unsupervised representations would not. It is also unclear to me the effect of the contrastive learning pre-selection (sec 7, last paragraph) in the disentanglement performance, how are the thresholds in Tab. 7 selected, how many contrastive models are discarded in this process and would disentanglement performance change if the same threshold was used for all factors or if this selection was not performed at all---as would be in practice where this factors are unknown.

Questions:
* It is not clear how the MI is computed for Fig. 3, I believe somehwere in the Appendix it said the distribution is approximated with a histogram, but some details (or a pointer to the details that are in Appendix) would be appreciated.
* I noticed the best validation FactorVAE metric across normalizations (0.916 in Tab. 4) is the same as the one reported in Tab.1, can you confirm this is by chance and not because the same dataset split was used for both.

Minor:
* abs: explored->explore
* Sec 1, p. 4: and Swav (Caron et al., 2020) etc -> , SWav (Caron et al., 2020), etc | and Swav (Caron et al., 2020).
* Sec 2, p. 2: "which maximize the information and the generated image"  seems to be missing words.
* Sec 2, last p.: drop the "or not"
* Sec 3.2: without notion otherwise -> unless otherwised stated
* Fig 2: add labels to the bars, e.g., "Contrastive representation" and "Ground truth factors"
* Tab 1, caption: contratstive -> contrastive

**Summary Of The Paper:**

Image representations learned using BYOL, a contrastive method, show better disentanglement than those learned with disentanglement-specific methods. Compares across models in toy datasets and one real-world dataset, CelebA.

**Summary Of The Review:**

Overall a nice paper that makes some hefty claims and as such, I would like to see more results, have some questions answered and discuss with other reviewers before I can recommend acceptance. Showing results hold for other contrastive methods would also go a great deal in showing the validity of them and increase the impact of this work but it might be outside its scope.

---

> ### Author Response · Authors · 2021-11-17
> **Response to Reviewer By54**
>
> Dear reviewer,
>
> Thank you so much for your reviewing and suggestions. Belows are our responses to your questions.
>
> **Q1. it would be interesting to see the complete set of results (probably in Appendix), namely Fig. 4 and Tab. 2 (or Tab. 1) for other datasets.**
>
> A1. We have made more complete evaluation results on all datasets. Please refer to the general response and the tables in Appendix B in the revised paper for details. Besides, to have more qualitative study of the disentanglement property of contrastive learning, we visualize the co-occurrence of mutual information of other datasets as we did in Fig.4 for dSprites. Please refer to Appendix C for the results and discussion. We believe all these provide more sense to help understand the property we study in this paper.
>
> **Q2. Insight about why contrastive representations are disentangled while supervised/unsupervised representation would not.**
>
> A2 First, we would like to clarify that the other unsupervised representations or supervised representations might also have disentangling properties, for example the previously studied VAE variants and GAN variants. In this paper, we focus on the contrastive learning’s disentanglement properties. Although we don’t have a thorough theory for why this happens, here is our preliminary hypothesis. Assuming the representation vector for two views x1 and x2 are f(x1) and f(x2). Since x1 and x2 are two augmentations of the same original image, very likely they will share most properties with each other, but only differ on very few properties. Thus, it might be more advantageous for the representation vector f(x1) and f(x2) to represent each semantically meaningful property in one dimension, since that would minimize the distance between f(x1) and f(x2), comparing to encoding one semantic change to all dimensions of f(x). We admit that this is rather preliminary, but we will add it to our main paper to provide some intuitions. We hope our empirical findings can inspire people to develop a more rigorous theory for it.
>
> **Q3. The effect of model pre-selection by factor prediction accuracy.**
>
> A3. The pre-selection by factor prediction accuracy is a sanity check to make sure models are not collapsed or learn not informative enough representations.
>
> (1) The threshold for each factor is determined by the performance of a well-trained baseline betaVAE model.
>
> (2) We did not use the pre-selection to pick good models. In fact, except for the model trained with instance normalization which collapses, no model in our experiments is discarded. The accuracy threshold also helps us to set an epoch to stop the training and make fair comparisons. For example, the BYOL models are always trained for 15 epochs on dSprites.
>
> (3) It is hard to set the threshold the same for all factors because some factors, such as orientation on dSprites which is defined ill-posed, are more difficult to predict. If we set the threshold uniformly low, e.g., 0.5 for all factors, it might motivate us to choose not fully trained models whose representation is not as informative as it can be.
>
> (4) The pre-selection helps us to find a good set of hyper-parameters. We describe it in the paper for completeness. However, in hindsight, the hyper-parameters are not too different from dataset to dataset. Thus, without pre-selection, the result will not change much.
>
> **Q4. How MI is computed.**
>
> A4. In appendix A.4 of the original paper version, we describe the details of MI calculation, "For Mutual Information computation, the used discretizer function is the histogram discretizer and the number of bins in discretization is 20". We will put a pointer to it in the corresponding positions of the main text in future version.
>
> **Q5. The best validation FactorVAE score (0.916) in Table 1 and  (the previous) Table 4.**
>
> A5. These two numbers are for the same model because the default setup of the used model is with group normalization (explained in Appendix A.1) which is different from the default version of BYOL. In Table 1 and the previous Table 4 (now Table 3), we report it twice to compare with different things. We will describe the default implementation earlier in the main text to avoid such confusion.
>
> **Q6. Minor issues about writing.**
>
> A6. Thank you very much for the detailed review. We have improved them in the revised version.
>
> Thank you,
>
> Authors

---

### Author Response · Authors · 2021-11-17
**General Responses**

We thank all reviewers for the time in reviewing this paper and the suggestions and questions which are very helpful for us to improve this work. We would like to make **a brief conclusion of this paper** here:

Given the disentanglement of representation is desirable as it provides good property on generalizability and interpretability. And it becomes even more valuable in unsupervised learning.  Having noticed it and the recent rise of contrastive learning, we conducted some empirical study on whether contrastive learning can learn disentangled representations. And what would influence the disentanglement learned by contrastive learning.

This paper mainly reports some empirical findings on learning disentangled representation by contrastive learning without negative samples. Given some previous works have noticed that contrastive learning can encourage feature disentanglement with contrast to large amounts of negative samples [W1], we think our work is the first to suggest it without negative samples, which is critical in previous analysis. We adopted BYOL[G1] with some modification of critical components, such as normalization and data augmentation details. We benchmark it with previous methods, mostly generative models, on the popular standard disentanglement datasets. The results show it can achieve SoTA performance with respect to multiple metrics.

We recognized some downsides of this paper such as it does not provide theoretical analysis about how the observed phenomena happen. But we believe the observations reported are still valuable and the path we provide to reproduce it and the ablation study we make on how to influence the disentanglement would be helpful to future works on both empirical line and more theoretical analysis.

Suggested by reviewers, we made more detailed benchmarking. We include some methods to compare with, including SlowVAE[K1] and EBM[K2]. In the new version, we replace the scores from absolute number to percentage (e.g., 0.916->91.6). Please refer to the following tables. We also note that on some dataset, the MIG and SAP score is below state-of-art. This is because the BYOL approach learns a group-disentangled representation. Therefore, there are multiple dimensions in the representation denoting the same ground truth factor. MIG and SAP score computes the mutual information difference between the most related and second related dimension to each ground truth factor. Thus a group disentangled feature can have low MIG and SAP score, even after the dimension reduction. Please check the revised version of paper for results:

**Table1**. The mean performance on dSprites with the standard deviation

**Table7**. The mean performance on Cars3D with the standard deviation

**Table8**. The mean performance on SmallNORB with the standard deviation

**Table9**. The median performance on Shapes3D (as the mean performance of some methods are not available) and the standard deviation of some methods are reported

For other point-to-point feedback, please refer to the responses below to reviewers. By gathering the comments of reviewers, we are uploading a new version of paper but the responses are all regarding the original version unless specified.

**The changelog for the revised version of paper**:
1. We add more quantitative results on synthetic datasets. And we add the standard deviation to help understand more accurately the model performance. We also add more methods to compare to.
2. We rewrite the related works section to more completely introduce the related literature, including some from the ICA community and add some previously missing reference suggested by the reviewers.
3. We add more qualitative results to understand the disentanglement property by the contrastive learning. To be precise, we add the matrix of co-occurrence of mutual information for factors on SmallNORB, Cars3D and Shapes3D.
4. We rewrote some paragraphs to add pointers to detailed explanations in the appendix.
5. We adjust statements or add explanations to make them more precise.
6. We add more explanation for the factor prediction accuracy pre-selection and correct the threshold for SmallNORB factors which was reported by my mistake in the previous version. All threshold comes from the pretrained BetaVAE model from disentanglement_lib.
7. Other minor updates about writing.

we note that limited by the time we have, as we want to give feedback to the reviewers as fast as possible, we would continue to polish the paper details.

[W1] Wang, Tongzhou, and Phillip Isola. "Understanding contrastive representation learning through alignment and uniformity on the hypersphere."

[G1] Grill, Jean-Bastien, et al. "Bootstrap your own latent: A new approach to self-supervised learning."

[K1] Klindt, David, et al. "Towards nonlinear disentanglement in natural data with temporal sparse coding."

[K2] Khemakhem, Ilyes, et al. "ICE-BeeM: Identifiable Conditional Energy-Based Deep Models Based on Nonlinear ICA."

---

> ### Author Response · Authors · 2021-11-29
> **An updated version of Table 2 and Table 3**
>
> To more accurately describe the experiment results, we update the Table 2 and Table 3 in the paper.
>
> **Table 2:** we re-trained the baseline models to use the same three random seeds for all methods (seed=0,1,2, same as the results on other datasets). We can't report DCI scores here as the evaluation of it is time-intensive, we will update that in the future version. As the result of InfoGAN-CR is from their officially released checkpoint and we have no access to multiple checkpoints from different random seeds, we thus can not report the standard deviation for it.
>
> |             | BetaVAE | FactorVAE | MIG  | SAP  |
> | ----------- | ------- | --------- | ---- | ---- |
> | VAE         |      21.6 (0.3)   |      4.1 (0.3)     |    0.7 (0.3)  |  1.1 (0.2)    |
> | $\beta$-VAE    |    13.1 (4.3)    |   3.5 (0.5)        |   0.2 (0.2)   |    0.9 (0.1)  |
> | $\beta$-TCVAE  |     24.5 (3.5)    |     6.7 (2.2)      |    0.7 (0.2)  |   1.2 (0.7)   |
> | FactorVAE   |     23.5 (2.2)    |      7.6 (4.6)     |   0.6 (0.1)   |    0.6 (1.4)  |
> | DIP-VAE-I   |    21.1 (0.5)     |       4.8 (1.3)   |   0.2 (0.1)   |    1.0 (1.4)  |
> | InfoGAN-CR | 16.8 | 11.3 | 1.6 | 2.8 |
> | BYOL (Ours) |    **35.7 (4.4)**     |    **11.5 (0.9)**       |  **2.6 (0.5)**   |  **8.2 (8.0)**    |
>
> **Table 3:** we add the standard deviation to better measure the results.
>
> | w/o norm   | BN         | GN             | LN         | IN         |
> | ---------- | ---------- | -------------- | ---------- | ---------- |
> | 89.6 (0.5) | 85.0 (1.4) | **91.6 (0.8)** | 91.4 (0.4) | 19.1 (0.0) |

---

### Decision · Program_Chairs · 2022-01-20

**Decision:**

Reject

**Comment:**

The paper provides additional empirical evidence that self-supervised learning methods can help disentangling factors of variation in a dataset. That said, the paper can benefit from better framing and perhaps comparison with existing work (e.g., https://arxiv.org/abs/2102.08850 and https://arxiv.org/abs/2007.00810). Furthermore, the authors acknowledge that there was a bug in their code, which I believe should at least lead to softening the claims about group disentanglement. Accordingly, please consider revising the paper and re-submitting to other venues.